

# Temporal variability of observed and simulated gross primary productivity, modulated by vegetation state and hydrometeorological drivers

Jan De Pue[1], Sebastian Wieneke[2], Ana Bastos[3], José Miguel Barrios[1], Liyang Liu[4], Philippe Ciais[4], Alirio Arboleda[1], Rafiq Hamdi[1], Maral Maleki[5], Fabienne Maignan[4], Françoise Gellens-Meulenberghs[1], Ivan Janssens[5], and Manuela Balzarolo[5]

[1]Department of Meteorological and Climatological Research, Royal Meteorological Institute, Belgium
[2]Remote Sensing Center for Earth System Research, University of Leipzig, Leipzig, Germany
[3]Department of Biogeochemical Integration, Max Planck Institute for Biogeochemistry, Jena, Germany
[4]Laboratoire des Sciences du Climat et de l'Environnement, LSCE/IPSL, CEA-CNRS-UVSQ, Université Paris-Saclay, Gif-sur-Yvette, France
[5]Department of Biology, University of Antwerp, Antwerp, Belgium

**Correspondence:** Jan De Pue (jan.depue@meteo.be)

**Abstract.** The gross primary production (GPP) of the terrestrial biosphere is a key source of variability in the global carbon cycle. It is modulated by hydrometeorological drivers (i.e., shortwave radiation, air temperature, vapor pressure deficit and soil moisture) and the vegetation state (i.e., canopy greenness, leaf area index) at instantaneous to interannual timescales. In this study, we set out to evaluate the ability of GPP-models to capture this variability. 11 models were considered, which rely purely
5   on remote sensing data (RS-driven), meteorological data (meteo-driven, e.g., dynamic global vegetation models; DGVMs) or a combination of both (hybrid, e.g., light-use efficiency models; LUE). They were evaluated using in situ observations at 61 eddy covariance sites, covering a broad range of herbaceous and forest biomes.

The results illustrated how the determinant of temporal variability shifts from meteorological variables at sub-seasonal timescales to biophysical variables at seasonal and interannual scale. RS-driven models lacked the sensitivity to the dominant drivers at
10   short timescales (i.e., shortwave radiation and vapor pressure deficit), and failed to capture the decoupling of photosynthesis and canopy greenness (e.g., in evergreen forests). Conversely, meteo-driven models accurately captured the variability accross timescales, despite the challenges in the prognostic simulation of the vegetation state. Largest errors were found in water-limited sites, where the accuracy of the soil moisture dynamics determines the quality of the GPP estimates. In arid herbaceous sites, canopy greenness and photosynthesis were more tightly coupled, resulting in improved results with RS-driven models.
15   Hybrid models capitalized on the combination of RS observations and meteorological information. LUE models were among the most accurate models to monitor GPP across all biomes, despite their simple architecture.

Overall, we conclude that the combination of meteorological drivers and remote sensing observations is required to yield an accurate reproduction of the spatio-temporal variability of GPP. To further advance the performance of DGVMs, improvements in the soil moisture dynamics and vegetation evolution are needed.



## 1 Introduction

Within the global carbon cycle, the exchange of carbon via photosynthesis and respiration in the terrestrial biosphere represents one of the largest and most dynamic components. Roughly 130 PgC/y flows through the plant stomata for gross primary productivity (GPP), from the total 875 PgC stored in the atmosphere (Ciais et al., 2013; Friedlingstein et al., 2022). During the decade 2012-2021, $3.1 \pm 0.6$ PgC/y was captured in the net terrestrial biosphere sink (i.e., gross primary productivity - ecosystem respiration). With an interannual variability of 1 PgC/y, it is considered the most variable element in the global carbon cycle (Friedlingstein et al., 2022). Despite the substantial role of GPP in the global carbon cycle, quantifying this flux remains associated with large uncertainties (Anav et al., 2015).

The temporal variability of GPP is largely modulated by the vegetation state (i.e., canopy greenness, leaf area index, etc.), and hydrometeorological conditions (Beer et al., 2010; Delpierre et al., 2012; Anav et al., 2015; Baldocchi et al., 2018). Consequently, most GPP models rely on remotely-sensed (RS) observations of the vegetation, meteorological forcings, or a combination thereof (Xiao et al., 2019; Friedlingstein et al., 2022; Jung et al., 2020). The vegetation state can be observed via remote sensing, making it an attractive approach to estimate global GPP dynamics. Vegetation indices (VI), such as the normalized difference vegetation index (NDVI; Rouse Jr et al., 1974), enhanced vegetation index (EVI; Huete et al., 2002) or near-infrared reflectance of vegetation (NIRv; Badgley et al., 2017), are indicators of the presence of (green) vegetation. Given their robustness and the availability of relatively long timeseries, the potential of these VI as a (linear) proxy for GPP has been explored by various studies (Tucker et al., 1986; Xiao et al., 2019; Huang et al., 2019; Balzarolo et al., 2019). Advancing beyond this, machine learning methods have been used to better exploit the potential of optical RS data in the recent decade (e.g., FluxCom; Jung et al., 2020), and the potential of new RS proxies with a more direct link to photosynthesis has been established, e.g., solar-induced chlorophyll fluorescence (SIF; Frankenberg et al., 2011; Liu et al., 2017; Pickering et al., 2022). The challenge associated with these models is that the relation between vegetation state and photosyntesis can decouple due to other limiting factors, such as soil moisture, temperature, and shortwave radiation (Walther et al., 2016; Hu et al., 2022). Opposed to RS-driven models, dynamic global vegetation models (DGVM) are driven largely by meteorological forcings. They are process-based models in which the exchanges of energy, water and carbon between the terrestrial biosphere and the atmosphere are simulated in a mechanistic manner. These models allow to assess the terrestrial carbon assimilation in the global carbon budget, or to investigate historic and future trends under a changing climate (Friedlingstein et al., 2022). The key challenge in these highly complex models is the correct representation of all underlying processes, including the dynamics of the canopy (Sitch et al., 2015; Fatichi et al., 2019). The entangled nature of these processes, and the resulting disagreements in the model conceptualizations contribute to the large spread between these models and uncertainty associated with the land surface sink in Earth System Models (Haughton et al., 2016; Collier et al., 2018; Seiler et al., 2022).

In the frame of this study, hybrid models are models that rely on a combination of RS observations of the vegetation state and meteorological forcings. The light use efficiency (LUE) model, proposed by Monteith (1972) is one of the most elementary formulations. Thanks to its compatibility with RS observations and limited input requirements, this semi-mechanistic approach is widely used and available in many flavours and degrees of complexity (Pei et al., 2022). Examples include the MODIS MOD17



GPP product (Running et al., 2004) or the LSA SAF GPP product (Satellite Application Facility on Land Surface Analysis; Martínez et al., 2020). These models benefit from the complementary information in RS data and meteorological forcings, but remain sensitive to uncertainties associated with RS observations of dense vegetation and the incomplete representation of soil moisture stress (Stocker et al., 2018; Xiao et al., 2019; Bloomfield et al., 2023)

The impact of vegetation and hydrometeorological conditions on the temporal variability of GPP ranges from instantaneous to
interannual timescales (Stoy et al., 2009; Mahecha et al., 2010; Linscheid et al., 2020). As the available GPP models vary in architecture, in the representation of underlying processes (or absence thereof) and -eminently- in their forcings, their short-comings vary across biomes and temporal scales (Anav et al., 2015; Mahecha et al., 2010; Xiao et al., 2019). Depending on their application, models are required to give a good estimate of annual variability, response to climate extremes, or changes in phenology. In order to adequately capture these temporal patterns, the time-scale dependent sensitivity of GPP to its drivers
needs to be represented accurately (Delpierre et al., 2012; Linscheid et al., 2021). Model evaluation studies or intercomparison studies are in this regard generally restricted to a single model type (RS-driven, meteo-driven or hybrid), driver and/or timescale (Mahecha et al., 2010; Delpierre et al., 2012; Shao et al., 2015). Despite important efforts made in this domain, most notable with the International Land Model Benchmarking system (ILAMB; Collier et al., 2018), it remains currently largely unclear what the inter-model trade-offs are.

The overall objective of this study is to evaluate the ability of various modelling approaches (RS-driven, meteo-driven or hybrid) to capture the temporal variability of GPP. By comparing the simulations of GPP with in situ eddy covariance obser-vations, we aim to assess 1) their performance across a broad range of biomes and temporal scales and 2) their sensitivity to drivers of GPP (i.e., vegetation state and hydrometeorological conditions).

## 2   Materials & Methods

### 2.1   Test sites


The evaluation of the GPP models was performed using in situ observations from eddy covariance stations. Test sites were se-lected from the FLUXNET2015 dataset (Pastorello et al., 2020) and the ICOS '2018 drought initiative' dataset (Drought 2018 Team and ICOS Ecosystem Thematic Centre, 2019). It was ensured that the sites had a homogeneous land cover (within 1 km radius from the tower, assessed using Google Earth), which could be captured by the remote sensing products. Additionally,
the sites were required to have a minimum of 3 years of GPP data since 01/01/2007 (i.e., the start of the SIF timeseries). This resulted in a selection of 61 sites, listed in Tab. 1. The dataset contained 461 years worth of GPP data, in which evergreen needleleaf forests (ENF) and the mid-latitude temperature-driven biome (MidL_T; Papagiannopoulou et al., 2018) were dom-inantly represented.




| ID | Name | Period | PFT | HCB | ID | Name | Period | PFT | HCB |
|----|------|--------|-----|-----|----|------|--------|-----|-----|
| AU-ASM | Alice Springs | 2009 - 2013 | ENF | SubTr_W | ES-LM2 | Majadas del Tietar South | 2013 - 2018 | SAV | Trans_E |
| AU-Cpr | Calperum | 2009 - 2014 | SAV | Trans_W | FI-Hyy | Hyytiälä | 1995 - 2018 | ENF | Bor_WT |
| AU-DaP | Daly River Savanna | 2006 - 2013 | GRA | Trans_E | FI-Let | Lettosuo | 2008 - 2018 | ENF | Bor_WT |
| AU-DaS | Daly River Cleared | 2007 - 2014 | SAV | Trans_E | FI-Var | Värriö | 2015 - 2018 | ENF | Bor_E |
| AU-Dry | Dry River | 2007 - 2014 | SAV | Trans_E | FR-Fon | Fontainebleau-Barbeau | 2004 - 2014 | DBF | MidL_T |
| AU-How | Howard Springs | 2000 - 2014 | WSA | Trans_E | FR-Hes | Hesse | 2013 - 2018 | DBF | MidL_T |
| AU-Stp | Sturt Plains | 2007 - 2014 | GRA | Trans_E | FR-Pue | Puéchabon | 1999 - 2014 | EBF | Trans_E |
| AU-Tum | Tumbarumba | 2000 - 2014 | EBF | Trans_E | GF-Guy | Guyaflux (French Guiana) | 2004 - 2015 | EBF | Tropic |
| BE-Bra | Brasschaat | 1995 - 2018 | MF | MidL_T | IT-Cp2 | Castelporziano2 | 2011 - 2018 | EBF | Trans_E |
| BE-Lon | Lonzée | 2003 - 2018 | CRO | MidL_T | IT-SR2 | San Rossore 2 | 2012 - 2018 | ENF | Trans_E |
| BE-Vie | Vielsalm | 1995 - 2018 | MF | MidL_T | IT-SRo | San Rossore | 1998 - 2012 | ENF | Trans_E |
| BR-Sa1 | Santarém-Km67 | 2002 - 2012 | EBF | Tropic | NL-Loo | Loobos | 1995 - 2018 | ENF | MidL_T |
| CA-Gro | Ontario - Groundhog River | 2003 - 2015 | MF | Bor_T | RU-Fy2 | Fyodorovskoye dry spruce | 2014 - 2018 | ENF | Bor_WT |
| CH-Lae | Lägeren | 2003 - 2018 | MF | MidL_T | RU-Fyo | Fyodorovskoye | 1997 - 2018 | ENF | Bor_WT |
| CZ-BK1 | Bílý Kříž forest | 2003 - 2018 | ENF | MidL_T | SE-Deg | Degerö | 2000 - 2018 | WET | Bor_WT |
| CZ-Lnz | Lanžhot | 2014 - 2018 | MF | MidL_T | SE-Htm | Hyltemossa | 2014 - 2018 | ENF | MidL_T |
| CZ-RAJ | Rájec | 2011 - 2018 | ENF | MidL_T | SE-Lnn | Lanna | 2013 - 2018 | CRO | MidL_T |
| CZ-Stn | Štítná | 2009 - 2018 | DBF | MidL_T | SE-Nor | Norunda | 2013 - 2018 | ENF | MidL_T |
| DE-Geb | Gebesee | 2000 - 2018 | CRO | MidL_T | SE-Ros | Rosinedal-3 | 2013 - 2018 | ENF | Bor_WT |
| DE-Hai | Hainich | 1999 - 2018 | DBF | MidL_T | SE-Svb | Svartberget | 2013 - 2018 | ENF | Bor_WT |
| DE-Hte | Hütelmoor | 2008 - 2018 | WET | MidL_T | US-ARM | Southern Great Plains | 2003 - 2013 | CRO | MidL_W |
| DE-Kli | Klingenberg | 2003 - 2018 | CRO | MidL_T | US-Ha1 | Harvard Forest EMS (HFR1) | 1991 - 2013 | DBF | MidL_W |
| DE-Obe | Oberbärenburg | 2007 - 2018 | ENF | MidL_T | US-Me6 | Metolius Young Pine Burn | 2010 - 2015 | ENF | Trans_E |
| DE-RuS | Selhausen Jülich | 2010 - 2018 | CRO | MidL_T | US-MMS | Morgan Monroe State Forest | 1999 - 2015 | DBF | MidL_W |
| DE-RuW | Wustebach | 2009 - 2018 | ENF | MidL_T | US-SRC | Santa Rita Creosote | 2008 - 2015 | OSH | Trans_E |
| DE-Seh | Selhausen | 2006 - 2010 | CRO | MidL_T | US-SRG | Santa Rita Grassland | 2008 - 2015 | GRA | Trans_E |
| DE-Spw | Spreewald | 2009 - 2014 | WET | MidL_T | US-SRM | Santa Rita Mesquite | 2004 - 2015 | WSA | Trans_E |
| DE-Tha | Tharandt | 1995 - 2018 | ENF | MidL_T | US-UMB | UMich Biological Station | 2000 - 2015 | DBF | Bor_T |
| DK-Sor | Sorø | 1995 - 2018 | DBF | MidL_T | US-UMd | UMBS Disturbance | 2007 - 2015 | DBF | Bor_T |
| ES-Abr | Albuera | 2014 - 2018 | SAV | Trans_E | ZA-Kru | Skukuza | 1999 - 2013 | SAV | Trans_W |
| ES-LM1 | Majadas del Tietar North | 2013 - 2018 | SAV | Trans_E | | | | | |

**Table 1.** Selection of 61 FLUXNET/ICOS sites used in this study. Classification by plant functional type (PFT; evergreen broadleaf forest: EBF, evergreen needleleaf forest: ENF, deciduous broadleaf forest: DBF, mixed forest: MF, wetland: WET, grassland: GRA, open shrubland: OSH, savanna: SAV, woody savanna: WSA, cropland: CRO) and hydroclimatic biome (HCB; Boreal / Mid-Latitude / Transitional / Sub-tropical / Tropical + Energy / Water / Temperature-driven; Papagiannopoulou et al., 2018). Note: only data beginning from 2007 was used in this study. All sites with data until 2018 are taken from the ICOS 2018 drought initiative, data for the other sites was collected from the FLUXNET2015 dataset.

All data was pre-processed with the ONEFLUX pipeline (Pastorello et al., 2020). The observed net ecosystem exchange was partitioned in the ecosystem respiration and GPP components using the daytime fluxes and a constant friction velocity threshold across years (labeled as GPP_DT_CUT in the database). Depending on site data quality, the reference GPP



(GPP_DT_CUT_REF) or mean GPP (GPP_DT_CUT_MEAN) method was selected.

Daily data with a quality flag indicating poor gapfilling (QF < 0.1) were discarded in the analysis. It was ensured that the same

time periods were considered for all models at each site.

The test sites were classified per plant functional type (PFT; taken from the FLUXNET/ICOS IGBP metadata) and hydro-climatic biome (HCB; Papagiannopoulou et al., 2018), see Tab. 1. The distribution of the sites across PFT and HCB is visualized in supplement section A. Seven PFT-HCB classes were selected for extra detailed analysis, given their importance and/or data quantity: evergreen broadleaf forest in tropical biome (EBF-Tropic), deciduous broadleaf forest in mid-

latitude temperature-driven biome (DBF-MidL_T), evergreen needleleaf forest in boreal water-temperature driven biome (ENF-Bor_WT), evergreen needleleaf forest in mid-latitude temperature-driven biome (ENF-MidL_T), evergreen needleleaf forest in transitional energy-driven biome (ENF-Trans_E), savanna in transitional energy-driven biome (SAV-Trans_E) and croplands in mid-latitude temperature-driven biome (CRO-MidL_T).

## 2.2 Meteorological data

Incoming short-wave radiation, long-wave radiation and precipitation data, required by the meteo-driven and hybrid GPP models, were taken from the half-hourly tower observations. Due to large gaps in the atmospheric humidity timeseries, ERA5 was used as an alternative source for air temperature, atmospheric humidity, wind speed, and atmospheric pressure (Hersbach et al., 2020). It was verified that the impact of the use of ERA5 instead of local observations was limited for these variables (not

shown here). The forcing from ERA5 (hourly resolution) was linearly interpolated to match the 30 minute temporal resolution from the tower observations. The atmospheric $CO_2$ concentration was taken from the TRENDY timeseries (Sitch et al., 2015, https://sites.exeter.ac.uk/trendy).

### 2.3 Remote sensing data

The simplest models considered were the linear regressions based on remote sensed proxies of GPP, including VI and SIF.

Remote sensing data was gathered from SPOT Vegetation+PROBA-V (SPV) for each tower location (the nearest pixel). Derived from this, the normalized difference vegetation index (NDVI), the enhanced vegetation index (EVI) and near infrared of vegetation (NIRv) according to Tucker (1979),Huete et al. (2002) and Badgley et al. (2017):

$$NDVI = \frac{R_{770-800} - R_{630-670}}{R_{770-800} + R_{630-670}} \qquad (1)$$

$$EVI = 2.5 \frac{R_{770-800} - R_{630-670}}{R_{770-800} + 6 * R_{630-670} + 7.5 * R_{460-475} + 1} \qquad (2)$$

$$NIRv = NDVI * R_{770-800} \qquad (3)$$



where $R$ is the reflectance between the wavelengths in the subscript (in nm). Wavelength 770-800 nm was used for the NIR reflectance, 630-670 nm for red reflectance, and 460-475 nm for blue band reflectance.

Additionally, the canopy structure-related near-infrared reflectance of vegetation multiplied by incoming sunlight (NIRvP) was included (Dechant et al., 2022). It was calculated as follows:

$$\text{NIRvP} = \text{NIRv} * \text{PAR} \qquad\qquad (4)$$

where PAR is the daily mean photosynthetically active radiation, derived from the in-situ incoming shortwave radiation observations.

For remotely sensed SIF data, we relied on the downscaled GOME-2 SIF product by Duveiller et al. (2020), given the coarse spatial resolution of the GOME-2 SIF product (>40 km) and the limited available timeseries of TROPOMI (starting in april

2018). The downscaling procedure involves a LUE methodology, involving NIRv, normalised difference water index (NDWI; Gao, 1996) and land surface temperature (LST) data from MODIS.

## 2.4 GPP models

A range of models to estimate GPP was selected, representing RS-driven, meteo-driven and hybrid approaches. An overview

is given in Tab. 2.

| Model | Method | | | Forcing | | | Reference |
|---|---|---|---|---|---|---|---|
| | | | | RS data | SWrad | Other meteo | |
| NDVI | Empirical | QR | RS-driven | SPV | - | - | This study |
| EVI | | QR | | SPV | - | - | This study |
| NIRv | | QR | | SPV | - | - | This study |
| SIF | | QR | | GOME-2* | - | - | This study |
| FluxCom$_{RS}$ | | ML | | MODIS | - | - | Jung et al. (2020) |
| NIRvP | | QR | Hybrid | SPV | in situ | - | This study |
| FluxCom$_{RSMet}$ | | ML | | MODIS | ERA5 | ERA5 | Jung et al. (2020) |
| MOD17 | Mechanistic | LUE | | MODIS | GEOS5 | GEOS5 | Running et al. (2004) |
| LSA SAF | | LUE | | CGLS | in situ | in situ + ERA5 | Martínez et al. (2020) |
| ISBA | | DGVM | Meteo | - | in situ | in situ + ERA5 | Delire et al. (2020) |
| ORCHIDEE | | DGVM | | - | in situ | in situ + ERA5 | Krinner et al. (2005) |

**Table 2.** Overview of the RS-driven, hybrid and meteo-driven GPP models used in this study. The following modelling methodologies are used: quantile regression (QR), machine learning (ML), light use efficiency models (LUE) and dynamic global vegetation models (DGVMs). The remote sensing (RS) sources are SPOT-Vegetation + PROBA-V (SPV), GOME-2, MODIS and Copernicus global land service (CGLS) products. The short-wave radiation (SWrad) and other meteorological data were obtained from in situ tower observations, ERA-5 and GEOS5. *The SIF data from GOME-2 was the downscaled product from Duveiller et al. (2020), using NIRv, NDWI and LST from MODIS.





**RS-based regression models**

The simplest models considered were the linear regressions based on remotely sensed proxies of GPP. A robust linear regression model of the RS data versus the daily GPP was constructed using quantile regression (Koenker and Hallock, 2001). The complete dataset was used to obtain a model for each proxy. The use of daily or 16-day average GPP did not have a strong

impact on the results. Only NIRvP, which used in situ incoming shortwave radiation observations, had a significantly steeper slope using the daily resolution GPP. More details are given in supplement section A.

Note that the training data used here was also used in the evaluation of the model performance. Furthermore, most models in this study were directly or indirectly trained with data from eddy covariance towers (e.g., FluxCom (Jung et al., 2020), ORCHIDEE (Friend et al., 2007), ...). Consequently, it was not possible to ensure an independent validation of the models.

To minimize impact on the study results, the evaluation was largely based on metrics that are not impacted by the slope of the linear regression (correlation and phenology analysis, see further below). Absolute errors and bias of the models were not evaluated in this study. Additionally, the robustness of the regression was verified by performing the regression 20 times using a random subset of 50% of the tower sites (see supplement section A). The regression for NDVI had the largest uncertainty, where the coefficient of variation of the slope was 9%. For the other proxies, this was around 4-5%. With this result, the quan-

tile regression was found to be robust and independent of the training data subselection. The impact of the shared data in the training and evaluation phase on the results is thus assumed to be limited.

**Machine learning models**

The FluxCom dataset consists of up-scaled FLUXNET observations, using machine learning, remote sensing data and meteo-

rological data (Jung et al., 2020). In this study, we considered the FluxCom$_{RS}$ GPP product (0.0833° grid, 8-daily resolution), which relies on MODIS observations, and the FluxCom$_{RSMet}$ GPP product (0.5° grid, daily resolution), which incorporates supplementary ERA5 meteorological data. Notably, a basic soil water balance model is used to derive the water availability index from the meteorological data, and ingest it in the FluxCom$_{RSMet}$ machine learning algorithm (Tramontana et al., 2016). For each tower location, the closest pixel was extracted from the database.


**Light use efficiency models**

As opposed to the pure RS data-driven methods described above, semi-mechanistic models have been developed, which incorporate meteorological forcings to estimate GPP. A widely-applied method, thanks to its compatibility with remote sensing observations, is the LUE model (Monteith, 1972). The core of this method is given in Eq. 5, where the plant productivity

depends on the absorbed photosynthetic active radiation (APAR) and a light use efficiency factor ($\epsilon$).

$$GPP = \epsilon APAR \qquad (5)$$



This approach forms the basis of the MODIS MOD17 GPP product (Running et al., 2004) and the LSA SAF GPP product (Martínez et al., 2020).

The algorithm behind MOD17 is a fairly simplistic formulation, where $\epsilon$ is linearly dependent on air temperature and vapor pressure deficit. Atmospheric forcings for this product are taken from the GMAO/NASA daily global meteorological reanalysis dataset, generated by GEOS-5. Soil moisture is not considered in the MOD17 model (Running et al., 2004). Conversely, in the LSA SAF model $\epsilon$ depends on the ratio between the actual and potential evapotranspiration. Consequently, the impact of soil moisture is indirectly considered.

For MOD17, the closest pixel was extracted for each tower site (MOD17 GPP is available at 1 km resolution with 8-daily interval). The LSA SAF GPP in this study was produced by executing the model for each site (As no global coverage or long timeseries were operationally available in the LSA SAF GPP product). The inputs for this model were LAI and the fraction of absorbed photosynthetic active radiation (FAPAR) from the Copernicus Global Land Service, and ERA5 + in situ meteorological forcings (see De Pue et al. (2022) for more details on the modelling approach).

**Dynamic global vegetation models**

DGVMs apply a largely mechanistic methodology to estimate GPP, and its temporal variability is driven exclusively by meteorological forcings. Here, ISBA (Delire et al., 2020) and ORCHIDEE (Krinner et al., 2005) were considered. ISBA is the component within Surfex v8.1 (SURFace EXternalisée), dedicated to the modelling of energy, water and carbon exchanges between the soil-vegetation-snow continuum and the atmosphere. The numerous processes involved in these exchanges (e.g., soil moisture dynamics, evapotranspiration, stomatal closure, canopy growth, canopy radiation transfer, etc.) are fully coupled. Similarly, ORCHIDEE is a well-established model for the simulation of vegetation in the context of earth system models. The version used here was the one that was prepared for the 6th coupled model inter-comparison project (CMIP6). Both DGVMs share a similar architecture, but rely on different formulations for the same processes (e.g., photosynthesis following Goudriaan et al. (1985) and Jacobs et al. (1996) in ISBA versus Farquhar et al. (1980) and Collatz et al. (1992) in ORCHIDEE) and differ in parametrization.

The models were configured to run with identical atmospheric forcing (constructed from ERA5 and in situ meteorological observations), identical land cover and prognostic vegetation growth. These models were run offline, and were not coupled to an atmospheric model. For more details on the LSM configuration and an in-depth evaluation of these models, see (De Pue et al., 2022).

## 2.5 Analysis

To evaluate the performance of the models to capture the temporal variability, the timeseries in the dataset were decomposed in 2 ways: 1) by separating the inter-site variability, seasonal variability and variability of seasonal anomalies, and 2) by separating daily, weekly, monthly, seasonal and interannual components with singular spectral analysis (SSA).



The performance at these timescales was evaluated by comparing the simulated variability (quantified by the standard deviation, $\sigma$) in observations and simulations, and by computing the Pearson correlation ($r$). Additionally, the covariance (cov) between GPP and its driver variables was used to assess the sensitivity of GPP to these variables. It was evaluated whether the models reproduce the observed patterns.

Finally, the accuracy of the simulated carbon phenology was evaluated, by comparing the timing of the simulated seasonal
GPP cycle with observations. Details on the methodology are given below.

**Inter-site, seasonal and seasonal anomalies**

The variability of the simulated and observed GPP was decomposed into the inter-site (i.e., spatial) component, seasonal component and the component associated with the anomalies. If we concatenate the GPP timeseries from all sites into one
array, we can decompose it as follows:

$$X_{all} = X_{site} + X_{seas} + X_{anom} \qquad (6)$$

where $X_{all}$ is the full dataset, $X_{site}$ contains the mean GPP of each site, $X_{seas}$ contains the mean seasonal cycle of each site (after subtracting the mean of the site), and $X_{anom}$ the resulting anomalies. The mean seasonal cycle was obtained by subtracting the timeseries mean, and computing the smoothed (20-day moving average) mean annual cycle. The accuracy of
the models to capture each of these components was evaluated using the metrics given further below.

**Singular spectrum analysis**

To assess the spectral nature of the modelled GPP anomalies, the observed and modelled signals were decomposed in 5 classes (daily, weekly, monthly, annual, and interannual) using Singular Spectrum Analysis (SSA, also referred to as Singular System
Analysis). SSA is a method which allows to decompose a signal into subsignals with specific spectral properties (Elsner and Tsonis, 1996; Golyandina et al., 2001). The approach used here was similar to the one proposed by Mahecha et al. (2007). The procedure can be summarized in two steps: the signal decomposition and the reconstruction of the subsignals. In the signal decomposition step, lagged windows of the original signal were stacked. This array was subsequently decomposed into its underlying orthogonal features by a PCA analysis. Resulting was a decomposition of the original series in elementary
subsignals, usually characterized by a simple oscillating feature.

Next, these elementary subsignals were binned according to their spectral properties to reconstruct subsignals with uniform spectral properties. In this study, similar bins as in the study by Mahecha et al. (2010) were used (see Tab. 3).





| Timescale | min - max period |
|-----------|------------------|
| Daily | <8 days |
| Weekly | 8 - 32 days |
| Monthly | 32 - 128 days |
| Annual | 128 - 512 days |
| Interannual | >512 days |

**Table 3.** Classification of the temporal scales in the SSA

As discussed by Mahecha et al. (2010), some elementary subsignals might contain features with mixed spectral properties. To avoid this, Mahecha et al. (2010) proposed a double pass procedure, where the SSA is applied again on the reconstructed subsignals. However, this procedure yielded limited improvements in this study. Instead it was found to be beneficial attribute a higher weight to the high frequency bins compared to the low frequency bins. This was achieved using weights proportional to the lower frequency limit of each bin. An example of the analysis is shown in Fig. 1.

The benefit of SSA compared to other spectral disaggregating methods (e.g., the Fourrier transformation), is that it is less sensitive to gaps in the dataset, and that it can handle datasets with a lower sampling frequency (e.g., the NDVI timeseries with 8-daily resolution). Consequently, datasets with lower sampling frequency have no signal at the daily timescale. The SSA was applied to the observed and simulated GPP, allowing evaluation at each timescale. The evaluation was performed using the metrics given below.

**Performance metrics**

The daily GPP estimations from the various models were compared to the observed GPP at the eddy covariance stations (Tab. 1). The variability of the (decomposed) timeseries was quantified using the standard deviation of the data ($\sigma$). The relative variance (rel.$\sigma^2$) of a timeseries component was calculated as

$$\text{rel.}\sigma^2 = \frac{\sigma^2_{comp}}{\sigma^2_{all}} \tag{7}$$

where $\sigma_{comp}$ and $\sigma_{all}$ are the $\sigma$ of the component and the full dataset, respectively. This calculation assumes all components to be independent (as the covariance is ignored). It was verified that the covariance of the components is negligible compared to the variance. Detailed results are given in supplement section A.

Furthermore, the performance of the models was quantified using the Pearson correlation $r$:

$$r = \frac{\sum^{n_o}(y^* - \overline{y^*})(y^o - \overline{y^o})}{\sqrt{\sum^{n_o}(y^* - \overline{y^*})^2 \sum^{n_o}(y^o - \overline{y^o})^2}} \tag{8}$$

where $y^*$ and $y^o$ are the predicted and observed values, $\overline{y}$ the mean of $y$ and $n_o$ the number of observations). Significant differences between the models were evaluated with the Wilcoxon signed-rank test.





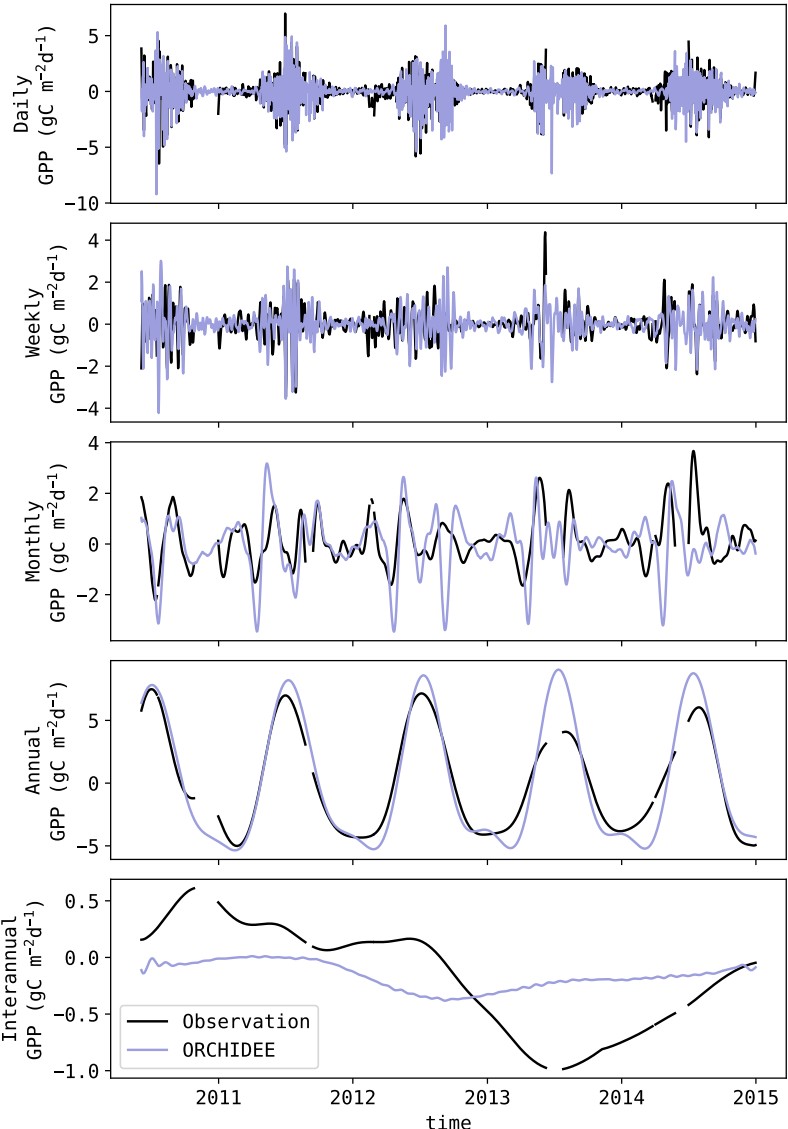

**Figure 1.** SSA decomposition of the observed GPP in DE-Spw and the simulation by ORCHIDEE.

Note that MOD17 or FluxCom$_{RS}$ are 8-day integrated GPP products, yet treated here as daily instantaneous products, analogous
to the other RS-based GPP products. Consequently, it can be expected that these GPP products will be less capable of estimating
the high-frequency anomalies.



**Driver variables**

Shortwave incoming radiation (SWrad; tower observation), air temperature at 2m (TA; tower observation), vapour pressure
deficit at 2m (VPD; tower observation) and surface soil moisture (SWC; ERA5) were selected as key hydrometeorological
drivers for GPP. Their impact at daily to interannual timescales was assessed by decomposing each timeseries using SSA and
calculating the covariance (cov) with GPP at each time scale. This was computed as:

$$\text{cov(x,y)} = \frac{1}{n_o} \sum_{i=1}^{n_o} (x_i - \overline{x})(y_i - \overline{y}) \tag{9}$$

where $x$ and $y$ represent two variables (e.g., GPP and SWrad). This analysis was performed for each site separately. The
similarity between the observed and simulated covariances was evaluated by comparing the median covariance across all sites,
and by computing the root mean square error (RMSE) between both. RMSE is computed as:

$$\text{RMSE} = \sqrt{\frac{\sum^{n_o} (y^* - y^o)^2}{n_o}} \tag{10}$$

where $y^*$ and $y^o$ are the predicted and observed values (covariances in this case). SWrad, TA and VPD were collected from
the tower meteorological observations. Given that no standardized soil moisture observations were available at each site, SWC
was taken from ERA5 (using the 0-7 cm layer) for each location.

As these drivers are not mutually independent, their covariance was evaluated for each HCB, and is given in supplement section
A. A positive covariance was found between SWrad, TA and VDP in most sites, and a negative covariance of these variables
with SWC. The covariances were the strongest at seasonal scale. Most HCB-classes showed similar behaviour, with some
exception in the Tropic and Trans_W biomes.


**Carbon phenology**

The (carbon) phenology in the timeseries was quantified by the timing of the start, maximum and end of the seasonal GPP cycle
(SOS, MOS and EOS). This was achieved by applying a smoothing operation (20 day rolling mean), followed by a threshold
procedure (Maleki et al., 2020; De Pue et al., 2022). In this procedure, the minima and maxima were used to delineate the
growing and senescent phase of the season. MOS was defined as the date when the maximum of the season is reached, SOS
and EOS were defined at the date where the growing or senescent phase crosses the threshold value $T$. $T$ was calculated for
each growing or senescent phase as $T = P_5 + 0.2(P_{95} - P_5)$, where $P_5$ and $P_{95}$ are the 5th and 95th percentile. If the seasonal
cycle was not pronounced enough $((P_5 - P_{95})/P_{50} < 0.2)$, the detected phenology was considered unreliable and ommitted.
The bias and accuracy of the phenology were evaluated by calculating the mean error (ME) and root mean square error (RMSE).




# 3 Results

## 3.1 Inter-site and seasonal variability

A comparison of the variability of GPP in observations and simulations is given in Tab. 4. The overall observed variability of $\sigma = 4.18 \text{ gC/m}^2/\text{d}$ was underestimated in all models, except LSA SAF. After decomposing the observed GPP dataset, the

inter-site variance represented 18% of the total variance, the seasonal cycles 62% and the anomalies 24% (the sum of these fractions is larger than 100% due to covariances, see supplement section A). This partitioning was not well represented in the NDVI, EVI and NIRv timeseries, where a large fraction of the variance ($> 30\%$) was attributed to the inter-site component, and a very small fraction ($< 12\%$) to the anomalies. In the NDVI observations, the inter-site variance was even larger than the seasonal variance. In SIF, the contribution of the spatial and seasonal components was reasonably accurate, but the relative

variance of the anomalies was too low (10%). The relative variance of the seasonal pattern was strongly overestimated in the FluxCom products ($\sim 80\%$), whereas the contribution of the anomalies was the lowest of all datasets ($\sim 5\%$). The closest match with the observed variance partitioning was found in NIRvP, MOD17, LSA SAF and the DGVMs. To ensure that these results were not affected by the temporal resolution of the timeseries, the same analysis was performed after downsampling to 10 day interval. This did not result in substantial changes of the variability or its partitioning (see supplement section A).


| | All | Inter-site | | Seasonal | | Anomalies | |
|---|---|---|---|---|---|---|---|
| Observation | 4.18 | 1.77 | 0.18 | 3.29 | 0.62 | 2.05 | 0.24 |
| NDVI | 2.10 | 1.46 | 0.48 | 1.33 | 0.40 | 0.74 | 0.12 |
| EVI | 2.95 | 1.69 | 0.33 | 2.25 | 0.58 | 0.90 | 0.09 |
| NIRv | 3.13 | 1.78 | 0.33 | 2.40 | 0.59 | 0.97 | 0.10 |
| FluxCom$_{RS}$ | 2.81 | 1.12 | 0.16 | 2.50 | 0.79 | 0.66 | 0.06 |
| SIF | 3.41 | 1.65 | 0.23 | 2.78 | 0.66 | 1.05 | 0.10 |
| NIRvP | 3.34 | 1.17 | 0.12 | 2.72 | 0.66 | 1.77 | 0.28 |
| FluxCom$_{RSMet}$ | 2.83 | 1.15 | 0.16 | 2.59 | 0.84 | 0.54 | 0.04 |
| MOD17 | 3.13 | 1.39 | 0.20 | 2.42 | 0.60 | 1.51 | 0.23 |
| LSA SAF | 4.83 | 2.24 | 0.21 | 3.68 | 0.58 | 2.38 | 0.24 |
| ISBA | 3.64 | 1.46 | 0.16 | 2.88 | 0.63 | 1.85 | 0.26 |
| ORCHIDEE | 3.68 | 1.34 | 0.13 | 3.18 | 0.75 | 1.75 | 0.23 |

**Table 4.** Standard deviation of the observed and simulated GPP ($\text{gC/m}^2/\text{d}$), decomposed in the inter-site, seasonal and anomalies (obtained after subtracting the spatial and seasonal component) components, and the fraction of the total variance (grey columns). This analysis done after grouping all sites together.

Depending on the land cover type, the variability and its partitioning between different components varied (Tab. 5). As expected, limited seasonal variability was observed in the EBF-Tropic sites ($\sigma_{season} = 0.68 \text{ gC/m}^2/\text{d}$), compared to DBF-MidL_T sites ($\sigma_{season} = 5.11 \text{ gC/m}^2/\text{d}$). Still, the variability of the anomalies of the tropical sites was comparable to that in other sites ($\sigma_{anom} \approx 2.00 \text{ gC/m}^2/\text{d}$). The CRO-MidL_T sites had the largest variability in the anomalies ($\sigma_{anom} = 3.43$

$\text{gC/m}^2/\text{d}$).





|  | All | Seasonal | | Anomalies | |
|---|---|---|---|---|---|
| EBF-Tropic | 2.25 | 0.68 | 0.09 | 2.15 | 0.91 |
| DBF-MidL_T | 5.11 | 4.79 | 0.90 | 2.01 | 0.17 |
| ENF-Bor_WT | 3.61 | 3.41 | 0.86 | 1.53 | 0.19 |
| ENF-MidL_T | 3.50 | 3.14 | 0.81 | 1.98 | 0.28 |
| ENF-Trans_E | 3.25 | 2.53 | 0.61 | 2.03 | 0.39 |
| SAV-Trans_E | 2.05 | 1.65 | 0.65 | 1.21 | 0.35 |
| CRO-MidL_T | 4.75 | 3.46 | 0.50 | 3.43 | 0.53 |

**Table 5.** Median standard deviation of the observed GPP per land cover class (gC/m$^2$/d), decomposed in the seasonal component and its anomalies. The fraction of the total variability is given in the grey columns.

The Taylor diagram of the modelled GPP and its seasonal anomalies is shown in Fig. 2. In terms of correlation, the DGVMs, LSA SAF and the FluxCom products achieved a distinctly better performance ($r > 0.83$, median for all sites), compared to the linear regression-based models (and MOD17). The NDVI-based model had the weakest correlation with observations ($r = 0.57$, median for all sites). The correlation of the simulated GPP was substantially reduced after subtracting the mean seasonal cycle. For NDVI, EVI, NIRv and SIF, $r_{anom}$ was smaller than 0.2 (median for all sites). LSA SAF and ISBA were the only models with $r_{anom} > 0.5$ (median for all sites). The performance of FluxCom to estimate the anomalies was similar to the NDVI-, EVI- and NIRv-based models. Though FluxCom$_{RSMeteo}$ achieved $r_{anom} = 0.43$ (median for all sites), the variability of the anomalies was strongly underestimated.

A notable difference emerged in the anomalies simulated with NIRvP and SIF. While both datasets showed a similar performance in the full GPP timeseries, SIF performed much poorer than NIRvP in the anomalies.

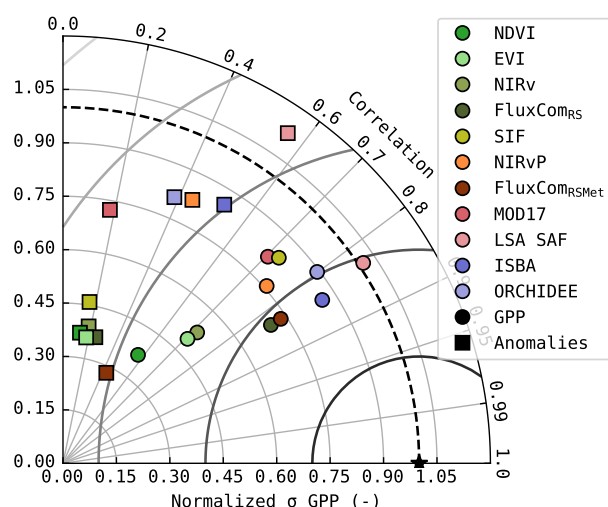

**Figure 2.** Taylor diagram of the simulated GPP (circles) and its seasonal anomalies (squares). Median of the metrics at all sites.





The RS-driven models, which relied purely on RS observation of the vegetation state, had a significantly lower $\sigma_{anom}$ (Wilcoxon $p < 0.05$) compared to the models that used meteorological forcing. This difference in performance was most pronounced in the forest sites (Fig. 3). In sites dominated by (water-limited) herbaceous vegetation, this was less the case; GPP

estimations based on simple greenness sensitive NDVI-, EVI- and NIRv-based models often even outperformed DGVMs.

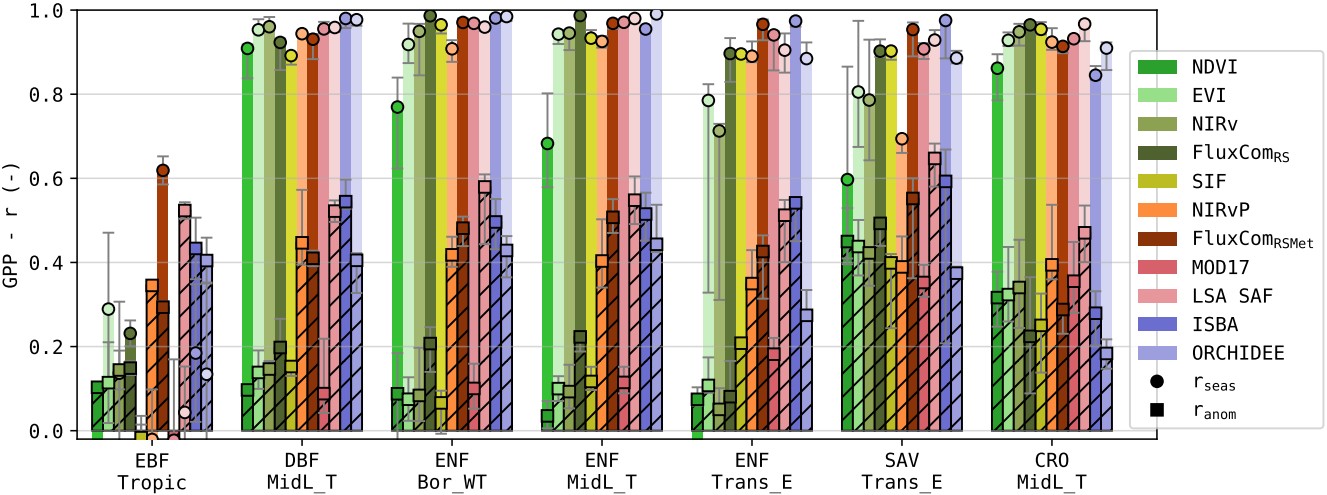

**Figure 3.** Pearson correlation of the modelled GPP and its anomalies, for sites in 7 PFT-HCB classes (see Tab. 1).

## 3.2 Timescale disaggregation

The variability of the timeseries after SSA decomposition is given in Tab. 6. In agreement with the variability of the seasonal GPP and its anomalies, the largest variability was explained by the annual timescale (77%, median for all sites). At daily, weekly and monthly timescale, the relative variance was roughly a tenfold smaller. The least variability was found for the

inter-annual scale (1%, median for all sites). More detailed results per land cover type are given in supplement section A. Most land covers followed the same pattern, with the exception of the EBF-Tropic sites, where seasonal variance was smaller than the variance at daily, weekly, monthly or even annual timescales.



| | All | Daily | | Weekly | | Monthly | | Annual | | Interannual | |
|---|---|---|---|---|---|---|---|---|---|---|---|
| Observation | 3.50 | 0.98 | 0.07 | 0.79 | 0.06 | 0.75 | 0.05 | 2.88 | 0.77 | 0.34 | 0.01 |
| NDVI | 1.20 | | | 0.21 | 0.06 | 0.47 | 0.33 | 0.65 | 0.57 | 0.05 | 0.00 |
| EVI | 1.60 | | | 0.21 | 0.04 | 0.52 | 0.24 | 1.10 | 0.70 | 0.07 | 0.00 |
| NIRv | 1.68 | | | 0.22 | 0.04 | 0.53 | 0.29 | 1.03 | 0.65 | 0.09 | 0.00 |
| FluxCom$_{RS}$ | 2.47 | | | 0.35 | 0.03 | 0.45 | 0.05 | 2.24 | 0.92 | 0.09 | 0.00 |
| SIF | 2.69 | | | 0.55 | 0.06 | 0.90 | 0.16 | 1.79 | 0.76 | 0.11 | 0.00 |
| NIRvP | 2.45 | | | 0.60 | 0.17 | 0.81 | 0.29 | 1.32 | 0.54 | 0.06 | 0.00 |
| FluxCom$_{RSMet}$ | 2.57 | 0.35 | 0.02 | 0.22 | 0.01 | 0.25 | 0.01 | 2.49 | 0.95 | 0.04 | 0.00 |
| MOD17 | 2.83 | | | 0.92 | 0.19 | 0.79 | 0.15 | 1.61 | 0.63 | 0.07 | 0.00 |
| LSA SAF | 3.59 | 1.42 | 0.16 | 0.82 | 0.06 | 0.63 | 0.03 | 3.09 | 0.72 | 0.17 | 0.00 |
| ISBA | 3.01 | 0.93 | 0.10 | 0.71 | 0.05 | 0.45 | 0.04 | 2.58 | 0.80 | 0.23 | 0.01 |
| ORCHIDEE | 3.13 | 0.72 | 0.06 | 0.57 | 0.03 | 0.73 | 0.06 | 2.70 | 0.85 | 0.15 | 0.00 |

**Table 6.** Standard deviation of the observed and simulated GPP ($gC/m^2/d$), decomposed in the different timescale components using SSA (median values for all test sites). The fraction of the total variability is given in the grey columns.

The RS-driven models underestimated the variance at all timescales, especially at annual scale. Furthermore, very limited
variability was found at the interannual scale, and the relative variance at monthly scale was overestimated in these models.
NDVI was the least suitable proxy to capture this variability, whereas the relative variance partitioning in SIF approximated
most closely the observations. Notably, the inclusion of PAR in NIRvP improved the GPP variability, but degraded the variance
partitioning across timescales.

The FluxCom products contained a too strong annual signal, and underestimated variability at other scales. The incorporation
of daily meteorological forcing in FluxCom$_{RSMet}$ added variability at the daily timescale, but reduced variability at weekly
and monthly timescale. The annual variability was approximated relatively well, but the variability at shorter timescales was
roughly threefold too small.

The variability across all timescales was best represented by the meteo-driven DGVMs (Tab. 6). There were minor differ-
ences between ORCHIDEE and ISBA, as the variance at daily and weekly scale was slightly more accurate in ISBA, and the
variance at monthly and annual timescale was more accurate in ORCHIDEE. This trend was confirmed in most land-covers
(see supplement section A). LSA SAF also estimated the variability reasonably accurate, but overestimated the daily variability.

The correlation of the simulations at these timescales is given in Fig. 4. Note that the strength of the signal at interannual
scale was relatively low (in observed and simulated GPP). Evaluating the correlation of this component should thus be done
with caution, as the SSA itself can induce errors of comparable magnitude (Mahecha et al., 2010). It is shown here, but not
discussed in detail.





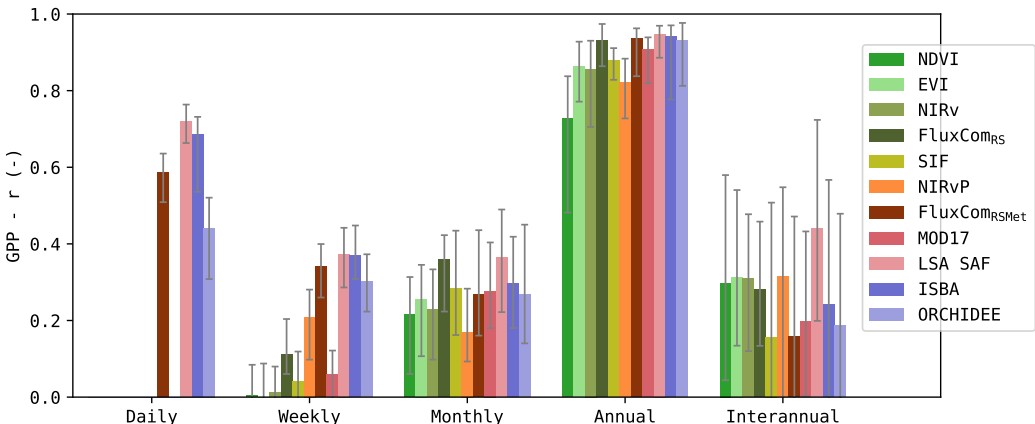

**Figure 4.** Pearson r across timescales after SSA decomposition; median score for all sites. Error bars indicate the 25-75 quantiles.

Most models had a good correlation with GPP at the annual timescale ($r > 0.80$, median for all sites), except the NDVI-based model. At monthly timescale, the correlation dropped to $r \approx 0.25$ for all models (median for all sites). At weekly timescale,
the models that relied solely on remote sensing observations were very poorly correlated to the observed GPP. Compared to these models, the models that included meteorological data achieved a significantly higher correlation (Wilcoxon $p < 0.05$). At daily scale, $r$ increased again. LSA SAF and ISBA achieved $r > 0.65$ (median for all sites) at this spectral range. Separating the results by PFT (Fig. 5) shows that the correlation at monthly and seasonal scale was generally larger for DBF-sites, compared to ENF-sites. At seasonal scale, this was most pronounced for the greenness-sensitive VI proxies (NDVI, EVI
and NIRv).





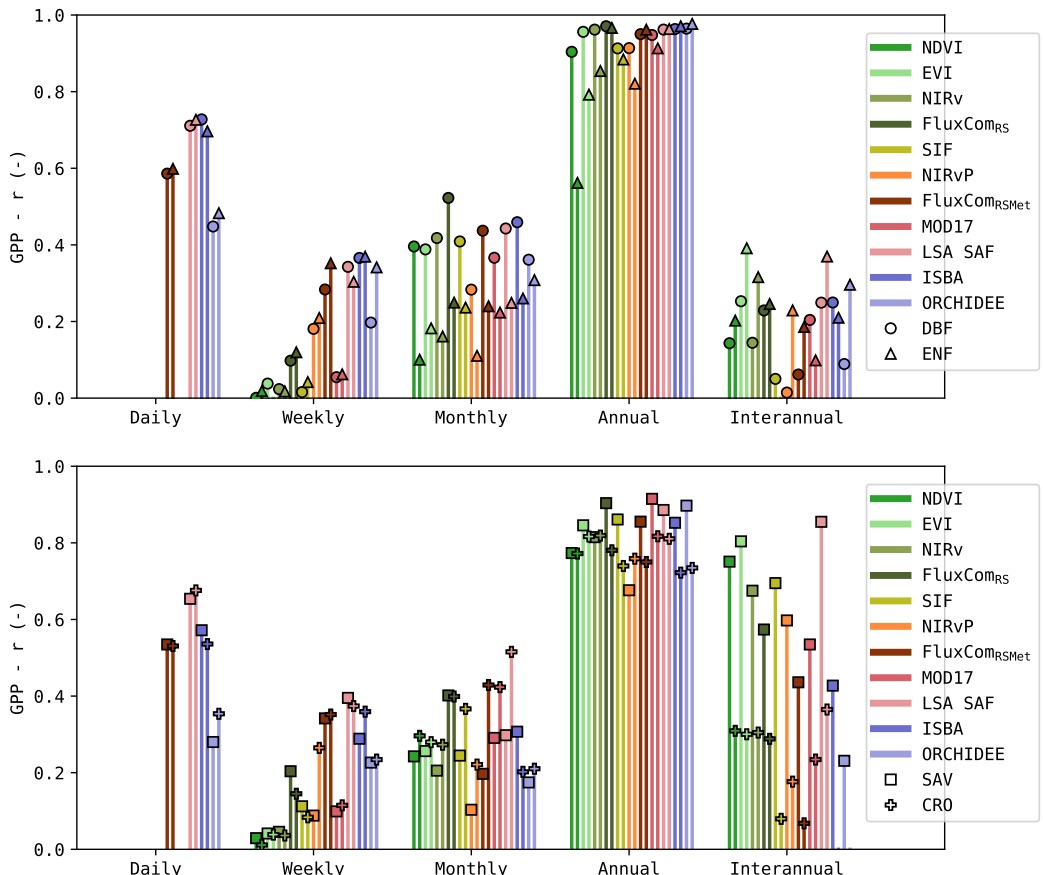

**Figure 5.** Pearson r in DBF, ENF, SAV and CRO across timescales

Dry-land sites, such as the SAV-sites, generally showed a higher correlation at the inter-annual scale for the RS-driven models. Not all models manage to capture these interannual patterns. For example, ORCHIDEE obtained only a very low correlation at this scale. Regardless, the interannual scale had only a minor contribution to the total variability.

In the CRO-sites, the RS-driven models had a significantly lower $r$ at weekly timescale, compared to the DGVMs (Wilcoxon $p < 0.05$, with the exception of NIRv vs ORCHIDEE and MOD17 vs ORCHIDEE). However, at monthly timescale, the RS-driven models had a higher $r$ than the DGVMs (significantly for SIF, FluxCom$_{RS}$ and MOD17), and at annual scale this trend persisted (significantly for EVI, NIRv, FluxCom$_{RSMet}$ and MOD17).

### 3.3 Drivers of GPP

Given the different performances across timescales, the covariance between the GPP and its key drivers (SWrad, TA, VPD and SWC) was evaluated. The observed and simulated covariances are shown in Fig. 6. These are the median covariances





for all sites. The covariance is impacted by the variance of the GPP estimates, as opposed to the Pearson correlation. For completeness, the latter is computed as well and is given in the supplement section A.



**Figure 6.** Covariance (median for all sites) of the simulated GPP and its drivers (SWrad, TA, VPD and SWC). Covariance based on observations are in the hashed bars. The colored barplots indicate the covariance in the simulations. Note that the covariance is shown using a symmetric logscale.

In the observations, all drivers had the highest covariance with GPP at the seasonal scale. SWrad and VPD had a stronger
covariance at daily scale compared to weekly and monthly scale, whereas TA had a slightly stronger covariance at weekly scale. The covariance between SWC and GPP was negative, indicating that GPP was smaller during wet rootzone soil moisture anomalies (and higher during dry anomalies). This was largely attributed to the negative covariance between SWC and the other drivers, as wet conditions are associated with periods of rain and cloudy weather (see supplement section A). The covariance between GPP and SWC was similar at daily, weekly and monthly scale. For all drivers and GPP, the interannual
signal was very weak, resulting in a negligibly small covariance.

Substantial differences in the observed correlations were found between different biomes, as highlighted in the plots of the



weekly, monthly and annual covariance (Fig. 7). For example, the covariance between SWrad and GPP at annual scale was very strong for most biomes, but it was very weak for EBF-Tropic (due to small variability of the GPP signal at this scale) and SAV-Trans_E sites (due to downregulation of photosynthesis by other constraining factors). Another clear trend was the shift
in covariance between SWC and GPP from negative in biomes where water is not a constraining factor (e.g., DBF-MidL_T) to waterlimited biomes (e.g., SAV-Trans_E).



**Figure 7.** Covariance of the simulated GPP and its drivers at weekly, monthly and seasonal scale. Covariance based on observations are in the hashed bars and gray bars highlight the deviation for a land cover from the overall average. The colored barplots indicate the covariance in the simulations.





The accuracy of the models to reproduce these patterns was quantified by RMSE (see supplement section A for detailed results). The RS-driven models generally had a very low sensitivity to all drivers at weekly and monthly scale. The covariances

at annual scale were underestimated as well. This can be attributed partly to the lower variance of the RS-driven GPP estimates at annual scale, but the Pearson r also indicated a too low sensitivity (see supplement section A). Conversely, the sensitivity of the meteo-driven models was generally more accurate. Some oversensitivity to the meteorological drivers was found in ISBA, whereas ORCHIDEE was generally among the most accurate models. The covariance with soil moisture was more accurate in ISBA than ORCHIDEE (e.g., RMSE at weekly, monthly and annual scale 10-30% more accurate)

The performance of the hybrid models was highly variable. LSA SAF was generally too sensitive to meteorological drivers, whereas MOD17 (also a LUE model) was too insensitive to all drivers (though more sensitive than the RS-driven models). The covariance of GPP with its drivers was generally most accurate in the FluxCom products. Their largest shortcoming was a too low sensitivity to SWrad at daily and weekly scale.

The dynamics in temperate DBF forest sites were reproduced fairy well by most models. The strong annual covariances were

represented well by all models. Even the RS-driven models had a relatively high covariance at this scale. At annual scale, the DGVMs and LSA SAF were most accurate in this biome (RMSE 3-4 fold lower than RS-driven models). In contrast, the high annual covariance was not represented well by the NDVI-,EVI and NIRv-based models in the ENF sites. The covariance between GPP and the drivers the drivers at annual scale was generally too weak. FluxCom and the DGVMs were more accurate (RMSE 4-5 fold lower than in VI-based models).

VPD and SWC were strong drivers for annual variability in the savanna sites. This was reproduced accurately by the RS-driven models, and ISBA. NIRvP, FluxCom$_{RSMet}$ and ORCHIDEE did not capture the annual covariance with VPD and SWC (RMSE for SWC and VPD 2-3 fold higher than ISBA, i.e., the most accurate model).

In the EBF-Tropic biome, all models had a too strong relation with the drivers at annual scale. Only in the FluxCom$_{RSMet}$ product, a resemblance with the observed annual relations was found. It was the only model with an accurate positive GPP-

SWC annual covariance for the EBF-Tropic sites.

The results for the FluxCom products highlight the importance of incorporating meteorological forcings in the GPP product. FluxCom$_{RSMet}$ was superior to FluxCom$_{RS}$ in the reproduction of GPP at different timescales. The coarser spatial resolution of FluxCom$_{RSMet}$ did not have a negative impact on the performance in this study.

This analysis gives a coarse estimate of the (linear) sensitivity of the simulated GPP to the drivers impacting GPP. Note that

many effects were not accounted for, including compound effects, legacy effects, or the impact of other constraining variables (e.g., LAI in the DGVMs).

### 3.4 Phenology

The accuracy of the simulated timing of the seasonal GPP cycle (start, max and end of season) is plotted in Fig. 8 (RMSE scores are calculated for every site individually). Generally, the simulations of SOS and EOS were generally less accurate in

the RS-driven models (RMSE SOS ≈ 30-38d, EOS ≈ 25-50d; except FluxCom$_{RS}$ ), compared to the meteo-driven models (RMSE SOS ≈ 24-28d, EOS ≈ 17-21d). The phenology in the NDVI-based model was the least accurate, which was largely



attributed to a bias in the timing, especially in the EOS ($\approx$ 50d delayed, see supplement section A). This bias was also observed in EVI and NIRv, but was smaller ($\approx$ 10d delayed). Notably, the most accurate simulations of SOS and EOS were obtained with FluxCom$_{RS}$, which purely relied on remote sensing observations.

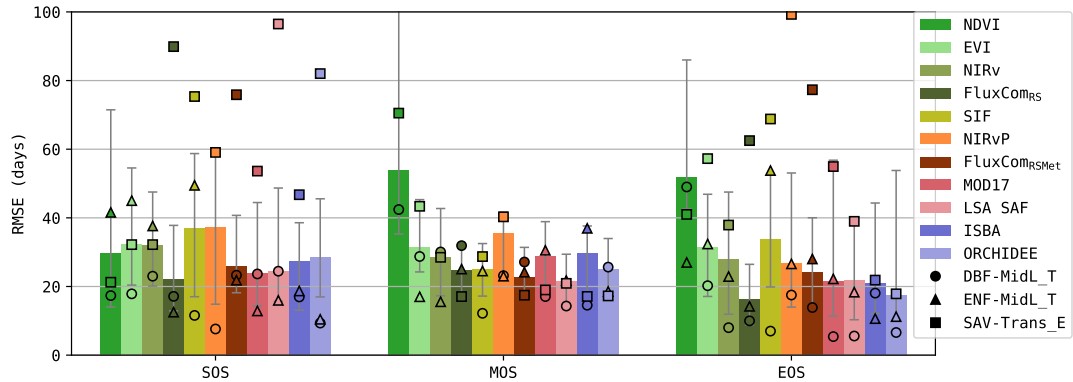

**Figure 8.** RMSE (per site) in the timing of the start, max and end of the seasonal GPP cycle (SOS, MOS and EOS). Bars show overall results (median for all sites), markers show separate results for three PFT-HCB classes.

To highlight differences between biomes, the mean annual cycle of DBF-MidL_T, ENF-MidL_T and SAV-Trans_E is plotted in Fig. 9 (the annual cycle of the other biomes can be found in the supplements). The DBF-MidL_T had a very distinct SOS around the 5th month of the year. The interannual variability of the observed GPP cycle was limited, compared to other biomes. Most models reproduced the phenology fairly accurately. In the NDVI-timeseries, an evident illustration of the so-called 'saturation effect' was observed, as the simulated GPP reached a plateau during mid-summer.

In the ENF-MidL_T biome, the coupling between canopy greenness and GPP was less strong than in DBF-MidL_T. Consequently, the meteo-driven and hybrid models were generally more accurate to simulate the timing of the GPP-cycle in this biome (RMSE SOS $\approx$ 10-18d, EOS $\approx$ 10-18d; see also Fig. 8) than the RS-driven models (RMSE SOS $\approx$ 35-50d, EOS $\approx$ 22-50d). Also note the delayed MOS in the ISBA simulations for this biome. This was largely associated with the delay in the prognostic LAI seasonal cycle (De Pue et al., 2022).

A strong variability of the annual GPP cycle was observed in the SAV-Trans_E biome (Fig. 9), making it very challenging to capture the timing of the GPP cycle accurately (Fig. 8). However, in these sites, a stronger coupling existed between GPP and the canopy greenness. At the SOS, a distinct difference between the RS-driven models and the meteo-driven models emerged. The RS-driven models were more accurate (RMSE SOS $\approx$ 20-30d for NDVI, EVI and NIRv), compared to the DGVMs (RMSE SOS $\approx$ 46-82d). In this biome, the inclusion of PAR in NIRvP resulted in a less accurate phenology compared to

NIRv. In NIRv, the reduced photosynthesis due to water-limiting conditions in the second half of the growing season was evident, whereas GPP remained high in NIRvP.





**Figure 9.** Annual GPP cycle in observations and models, for sites in the DBF-MidL_T, ENF-MidL_T and SAV-Trans_E biomes. The lines show the median cycle, and the shaded area shows the 25-75 percentile. Timeseries of sites located at the southern hemisphere were shifted by 6 months, to match with the annual cycle of sites in the northern hemisphere.



## 4 Discussion

The variability of GPP is largely modulated by the vegetation state (i.e., canopy greenness, leaf area index, etc), and hy-
drometeorological conditions. As indicated by Stoy et al. (2009), the relation of GPP to these factors shifts across timescales:
"Quantifying flux variability at longer time scales requires information on how ecosystems change in response to climatic
variability, rather than how they merely respond to climatic variability". In this study, we investigated how well the impact of
these factors is captured in RS-driven, meteo-driven and hybrid models.

### 4.1 Vegetation state

Depending on the biome, the vegetation state is tightly coupled (e.g., in water-limited herbaceous sites), more loosely coupled
(e.g., deciduous broadleaf forests) or completely decoupled (e.g., tropical evergreen broadleaf forests) to GPP (Hu et al., 2022).
Vegetation indices, such as NDVI, EVI and NIRv are effective proxies to track the vegetation state via remote sensing. They
have proven to be an effective, low-cost proxy for GPP in biomes with an evident coupling between canopy greenness and
photosynthesis (Xiao et al., 2019; Huang et al., 2019).

However, an important discrepancy was found between the RS observations and the observed GPP in the spatio-temporal par-
titioning of their variability. The inter-site variability of NDVI, EVI, NIRv and (to a lesser extent) SIF was substantially higher
than that of the GPP observations. Furthermore, the variability of the anomalies in the models was relatively small (see Tab 4).
This high inter-site variability indicated that there was a need to use land cover-dependent relations to estimate GPP from the
remotely sensed vegetation proxies. Several studies have confirmed that PFT-specific relations considerably improved the GPP
estimates from NDVI (Huang et al., 2019), EVI (Shi et al., 2017; Huang et al., 2019), NIRv (Badgley et al., 2019; Huang et al.,
2019) and SIF (Gao et al., 2021). FluxCom$_{RS}$ also relied on land cover data to estimate GPP from RS observations (Jung et al.,
2020), and captured the spatial and seasonal variability more accurately (see Tab. 4).

The biome-dependent relation between vegetation greenness and GPP was also evident in the seasonal cycle (Fig. 3) and in the
annual timescale (Fig. 5). For DBF and CRO biomes, the coupling between VI and GPP resulted in high correlations at these
timescales, whereas the decoupling in other biomes emerged. This was most pronounced in evergreen forest sites (ENF and
EBF), and the decoupling increased as the climate was increasingly water-limited (ENF-Bor_WT < ENF-MidL_T < ENF-
Trans_E, see Fig. 3 and Fig. 7). Opposed to herbaceous sites in the same arid biomes (e.g., SAV-Trans_E), the photosynthesis
downregulation in ENF sites was not translated into rapid changes in vegetation greenness.

As often reported, the decoupling of leaf phenology and carbon phenology was also poorly captured in the VI-based models.
This was most pronounced in the senescent phase, where photosynthesis halts, due to decrease in SWrad and TA, before canopy
greenness drops (Kong et al., 2020; Wang et al., 2020).

All VI were insensitive to the decoupling of canopy greenness and photosynthesis at seasonal timescale, but NDVI performed
significantly worse than EVI and NIRv in this respect. Saturation in dense canopies, background effects and atmospheric in-
fluences (Huete et al., 2002; Olofsson et al., 2008) likely explain the underestimated variability of the seasonal cycle in NDVI
timeseries, especially in forest biomes (illustrated in Fig. 9). Between EVI and NIRv, no substantial differences in performance



were found.

SIF is a more direct proxy for photosynthesis, and is therefore expected to capture the decoupling between vegetation greenness and GPP more accurately than VI (Duveiller et al., 2020; Pickering et al., 2022). However, SIF did not perform significantly better than EVI or NIRv at annual timescale (Fig. 4 and Fig. 5). Exceptions were the arid biomes, ENF_Trans-E and SAV_Trans-E, where SIF outperformed EVI and NIRv. It remains unclear to in what measure the downscaling processing is responsible for the moderate SIF scores. Future missions with high resolution SIF, such as European Spatial Agency's Earth Explorer - FLEX (FLuorescence EXplorer, due to be launched in 2025) will provide further insights (Duveiller et al., 2020).

The results with the VI-based models seemed to indicate that the remotely sensed observations of the vegetation state were insufficient to describe GPP in evergreen vegetation. However, FluxCom$_{RS}$ relied exclusively on these observations as predictors, and managed to capture GPP patterns in ENF. Furthermore, it produced the most accurate results regarding the GPP phenology. This product illustrated that, in combination with land cover information and non-linear relations, accurate estimates of GPP at seasonal timescale can be obtained from optical remote sensing (Tramontana et al., 2016).

Conversely, it is very challenging to accurately model the state of the vegetation without RS observations (Fatichi et al., 2019). In a detailed evaluation of the water, energy and carbon modelling in ISBA and ORCHIDEE, it was reported that the leaf phenology in ISBA and ORCHIDEE was delayed compared to observations and that it failed to capture the observed seasonal variability. De Pue et al. (2022) reported that these errors were strongly correlated to errors in GPP. Despite these inaccuracies, the performance of the DGVMs was generally better than the VI-based models. The dominant impact of meteorological forcings, and the decoupling of greenness and photosynthesis was captured accurately in the DGVMs.

Next to the complexity of plant physiology and biomass allocation, there can be a substantial impact of management practices (e.g., crop rotations, sowing and harvest in croplands ; Osborne et al., 2010). The lack of these practices in the configuration of the DGVMs in this study resulted also in a poorer performance of the monthly and annual-scale GPP in croplands (see Fig. 5). Observations of these practices in remote sensing contribute to a better performance in croplands with RS-driven models. At a global scale, the lack of an adequate description of land management contributes considerably to uncertainties associated with the global carbon cycle in earth system models (Friedlingstein et al., 2022).

In summary, based on the observed vegetation state, a coarse estimate of the annual-scale GPP can be made. However, vegetation indices and linear regressions are insufficient to capture the decoupling of greenness and photosynthesis due to other confounding factors. Information on the hydrometeorological conditions is needed to capture this variability in all biomes, even at seasonal scale.

## 4.2 Meteorological conditions

Meteorological conditions are the main drivers of variability of GPP at sub-seasonal scale (Stoy et al., 2009). At daily timescale, patterns were largely dominated by SWrad and VPD (see Fig. 6). The impact of TA was more pronounced at weekly and monthly scale (though still dominated by SWrad).

The RS-driven models had a very low performance to simulate these sub-seasonal patterns (Fig. 4). They had a temporal reso-



lution of 8-10 days, so the variability at daily timescale was absent. At weekly and monthly scale, they had nearly no sensitivity to the driver variables (Fig. 6). Consequently, the correlation of the anomalies was very weak in comparison to other models (Fig. 3).

NIRvP was the most simplistic approach to incorporate SWrad (as PAR) as key driver for photosynthesis (Eq. 4). Compared to NIRv (and SIF), NIRvP captured anomalies in GPP more accurately, in particular at the weekly timescale (Fig. 2, and Fig. 4).

Alternatively, light-use efficiency models ingest more meteorological variables, such as VPD and TA, in addition to SWrad and vegetation state variables. Consequently, the quality of the simulated GPP strongly depended on the quality of the meteorological forcings. The MOD17 product relied on the coarse GMAO/NASA reanalysis dataset for the meteorological forcing, and failed to achieve a better performance than the VI-based models (Fig. 2). The LSA SAF GPP model, here forced by in situ SWrad observations, excelled in the simulation of temporal variability at all timescales and in all domains. Although there

were other factors that impact the performance (e.g., the incorporation of soil moisture stress, which was absent in MOD17), the difference in SWrad forcings likely contributed substantially to the difference in performance, given the sensitivity of the models to SWrad and the quality of SWrad in reanalysis products (Anav et al., 2015; Urraca et al., 2018; Zheng et al., 2018). The incorporation of meteorological forcings in FluxCom$_{RSMet}$ improved the algorithm's ability to capture the anomalies, compared to FluxCom$_{RS}$ (Fig. 2). This was most evident in forest sites (Fig 3), though the improvement was restricted to the weekly

timescale (Fig. 4). Still, despite the introduction of meteorological variables, the variance of the anomalies remained strongly underestimated (Tab 4).

In contrast, the meteo-driven DGVMs represented the variability of GPP accurately across timescales. A significant difference between ISBA and ORCHIDEE was found in the performance at daily timescale (Fig. 4). The superior performance of ISBA at this timescale, seemed to be originating from a more accurate sensitivity to SWrad than ORCHIDEE (Fig. 6). Conversely,

the sensitivity to atmospheric drivers at weekly and monthly timescales was more accurate in ORCHIDEE, whereas ISBA was generally oversensitive (Fig. 6 and 5). Though the performance of ORCHIDEE to simulate GPP at these longer timescales was not superior (due to other confounding factors, e.g., soil moisture or LAI), ORCHIDEE is likely more accurate in assessing the impact larger meteorological anomalies, such as heat waves, on GPP. Further research, addressing the performance of the models under extreme conditions is needed to confirm this.


## 4.3 Soil moisture

At sub-seasonal scale, the RS-driven models demonstrated a big difference in performance between forest and herbaceous biomes. A substantially better performance was achieved in herbaceous sites (Fig. 3), where the coupling between vegetation greenness and GPP is much tighter than in forest sites Hu et al. (2022). The indirect observation of soil moisture stress in VI

allowed accurate sub-seasonal-scale modeling of GPP in these strongly water-limited biomes (AghaKouchak et al., 2015). In other biomes, the combination with a drought indicator is required to simulate GPP in such conditions (Maleki et al., 2022).

No downregulation due to soil moisture or temperature stress is considered explicitly in NIRvP. However, changes in light use efficiency are partly reflected in changes in the canopy structure (Xu et al., 2021). Consequently, NIRvP can yield similar





results than SIF, as demonstrated in the work by (Dechant et al., 2022). Regardless, in water-limited herbaceous sites (e.g.,
SAV-Trans_E), the sensitivity to soil moisture stress in NIRv was eliminated in NIRvP, due to a too high sensitivity to SWrad
(Fig. 7). An illustration of this lack of soil moisture stress downregulation was evident in the mean annual cycle of NIRvP,
where GPP was consistently overestimated during the dry season (Fig. 9). The downregulation was more accurately reflected
in the SIF model.

The seasonal GPP patterns in water-limited sites (e.g., ENF-Trans_E or SAV-Trans_E) were generally simulated less accurately
(see Fig. 3) in DGVMs, indicating that the soil moisture dynamics or the soil moisture stress response of the vegetation were
an important source of errors (Vereecken et al., 2019; Raoult et al., 2021; De Pue et al., 2022). In the arid biomes, differences
between ISBA and ORCHIDEE were most evident. The soil moisture dynamics and response to soil moisture stress in OR-
CHIDEE were demonstrated to be less accurate compared to ISBA in a previous study by De Pue et al. (2022).

## 4.4 Uncertainties

The in situ observation uncertainty may contribute to the disagreement between models and observations. The eddy covariance
observations are associated with site-dependent random errors due to instrumentation, stochastic nature of turbulence and vary-
ing footprint (Mauder et al., 2020). Additionally, the typical non-closure of the energy balance might indicate that the observed
carbon fluxes suffer from a similar bias (Gao et al., 2019), and there are significant uncertainties associated with the carbon
flux partitioning in the ONEFLUX preprocessing pipeline (Pastorello et al., 2020).

Though land cover homogeneity and data quality were criteria for site selection, the discrepancy between the spatial scale of
the in situ and remote sensing observations may contribute to the disagreement between observed and simulated GPP (Xie
et al., 2021). Furthermore, there is a representation bias in the selection of test-sites used here. There are limited sites included
from South America, Africa and Asia. Consequently, some of the results reported here might be biased due to the dominant
representation of (needleleaf) forest sites in temperate climates.

Lag effects of the drivers were not investigated in the frame of this study. Generally, it is mainly precipitation which leads
to time lag effects (Papagiannopoulou et al., 2017), but that effect was largely accounted for by considering soil moisture.
However, severe drought extremes can have a legacy effect, with a substantial impact on the inter-annual variability of GPP in
terrestrial ecosystems (Bastos et al., 2020). These effects fall out of the scope of this study.

The interannual variability in the SSA-decomposed timeseries was relatively small, in agreement with the results of Mahecha
et al. (2010). Given the associated uncertainty, and relatively short timeseries in most sites, interpretation of the results at these
timescales should be done with caution. In savanna biomes, there was an indication that RS-driven models captured the inter-
annual variability better than meteo-driven models (see Fig. 5). In other biomes, the interannual correlation was very weak.

This study evaluated the ability of the models to capture the variability in GPP. It relied on analysis of the variance, the Pearson
correlation, and metrics for phenology. The absolute errors were not evaluated here. These results give no guidance on the bias
or accuracy of the simulated GPP itself.



## 5 Conclusions

The temporal variability of GPP is modulated by vegetation state and hydrometeorological factors, operating at instantaneous
to interannual timescales. In this study, we set out to evaluate the ability of GPP-models to capture this variability. 11 models
were considered, encompassing remote sensing-driven models (e.g., NDVI regression, SIF, FluxCom$_{RS}$ ), meteo-driven models
(i.e., ISBA and ORCHIDEE DGVMs), and hybrid models that combined both inputs (e.g., FluxCom$_{RSMet}$ or LUE algorithms,
such as MOD17 and LSA SAF). They were evaluated using in situ observations of GPP at 61 eddy covariance sites, covering
a broad range of biomes. The analysis comprises decomposition of the signal in daily to inter-annual timescales, covariance
with driver variables and phenology.

The results illustrated how the determinant of temporal variability shifts from meteorological variables at sub-seasonal timescales
to biophysical variables at seasonal and interannual scale. Consequently, shortcomings were accordingly associated with RS-
driven and meteo-driven models. To capture the full range of variability accurately, RS-driven models lack the sensitivity to the
dominant drivers at short timescales, i.e., SWrad and VPD. Furthermore, they failed to capture the decoupling of photosynthe-
sis and canopy greenness in evergreen vegetation or during senescence. Conversely, meteo-driven models accurately captured
the variability accross timescales. Though the progonostic simulation of the vegetation state remains elusive, the seasonal pat-
terns in GPP are accurately reproduced.

Important challenges remain in the simulation of soil moisture and the response of vegetation to soil moisture stress, illustrated
by the poorer performance of the DGVMs in water-limited sites. RS-driven models captured the GPP anomalies accurately in
these sites, as they were characterized by a tight coupling of vegetation greenness.

Hybrid models capitalized on the combination of RS observations and meteorological information. The simple inclusion of
PAR in NIRvP was beneficial to capture the variability of GPP at all timescales. LUE models were among the most accurate
models to monitor GPP across all biomes, but large differences between MOD17 and LSA SAF illustrated their sensitivity to
the quality of the meteorological forcings used.

Overall, we conclude that the combination of meteorological drivers and remote sensing observations are needed to yield an
accurate reproduction of the spatio-temporal variability of GPP. To further advance the performance of DGVMs, improvements
in the soil moisture dynamics and vegetation evolution are needed.

*Code and data availability.* The dataset is published at https://www.zenodo.org/ (DOI: `10.5281/zenodo.7928514`). It contains GPP
from all sources + in situ radiation, temperature, vapor pressure deficit and ERA5 soil moisture. The scripts used in this study are freely
available upon request to the authors.



# Appendix

## Validation Sites

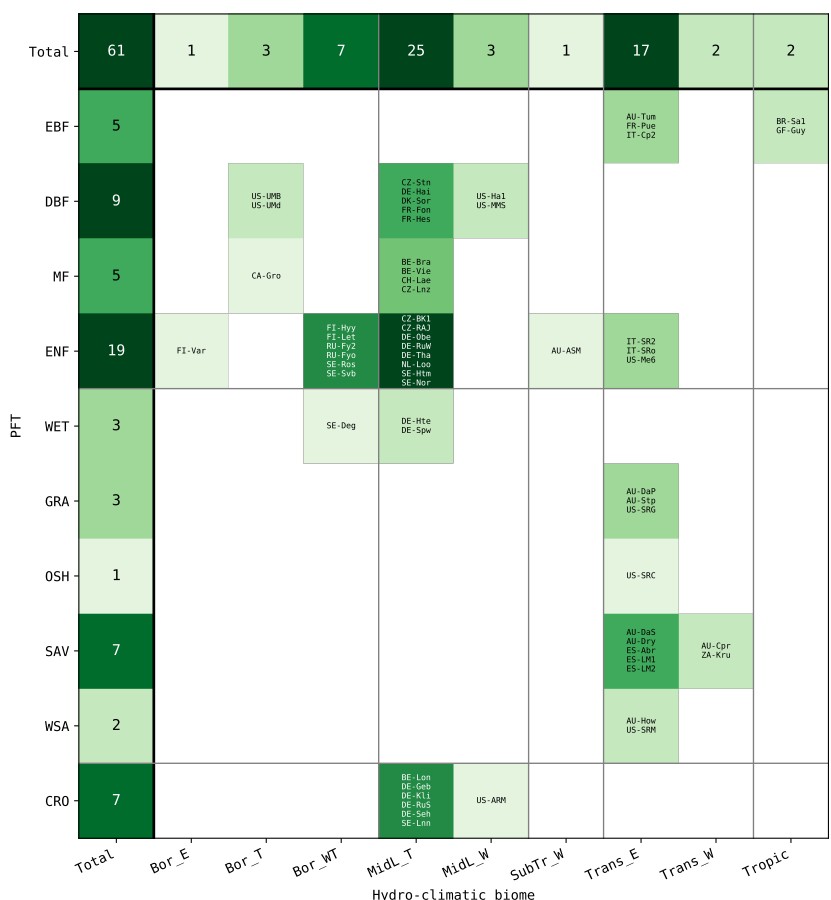

**Figure A1.** Distribution of the selected testsites across PFT and HCB.





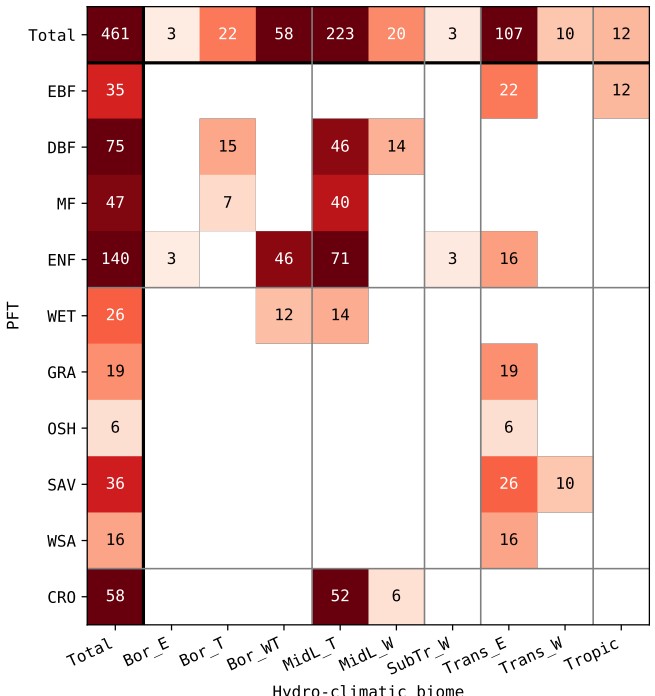

**Figure A2.** Total length of GPP timeseries available (years), aggregated per PFT and HCB.




**Quantile regression**



**Figure A3.** Evaluation of the robustness of the quantile regression between RS observations and tower GPP. The regression was done for daily data (red) and 16-day aggregated data (blue). For each approach, the opaque line shows the mean of 20 regressions using a 50% subsample of the dataset (transparent lines).





595 **Impact temporal resolution**

|  | All | Inter-site |  | Seasonal |  | Anomalies |  |
|---|---|---|---|---|---|---|---|
| Observation | 4.16 | 1.75 | 0.18 | 3.17 | 0.58 | 2.01 | 0.24 |
| NDVI | 2.11 | 1.46 | 0.48 | 1.32 | 0.40 | 0.72 | 0.12 |
| EVI | 2.95 | 1.70 | 0.33 | 2.23 | 0.57 | 0.86 | 0.08 |
| NIRv | 3.13 | 1.79 | 0.33 | 2.38 | 0.58 | 0.92 | 0.09 |
| NIRvP | 3.34 | 1.17 | 0.12 | 2.57 | 0.59 | 1.75 | 0.27 |
| SIF | 3.42 | 1.63 | 0.23 | 2.78 | 0.66 | 1.06 | 0.10 |
| FluxCom$_{RS}$ | 2.82 | 1.12 | 0.16 | 2.48 | 0.78 | 0.67 | 0.06 |
| FluxCom$_{RSMet}$ | 2.83 | 1.15 | 0.17 | 2.52 | 0.79 | 0.52 | 0.03 |
| MOD17 | 3.13 | 1.36 | 0.19 | 2.38 | 0.58 | 1.47 | 0.22 |
| LSA SAF | 4.83 | 2.23 | 0.21 | 3.59 | 0.55 | 2.30 | 0.23 |
| ISBA | 3.61 | 1.45 | 0.16 | 2.76 | 0.58 | 1.80 | 0.25 |
| ORCHIDEE | 3.68 | 1.34 | 0.13 | 2.95 | 0.64 | 1.68 | 0.21 |

**Table 1.** Standard deviation of the observed and simulated GPP ($gC/m^2/d$), decomposed in the inter-site, seasonal and anomalies (obtained after subtracting the spatial and seasonal component) components, and the fraction of the total variance (grey columns). This analysis done after grouping all sites together. A downsampling to 10-daily resolution was performed prior to this analysis.



## Variability

| | | All | Daily | | Weekly | | Monthly | | Annual | | Interannual | |
|---|---|---|---|---|---|---|---|---|---|---|---|---|
| Obs | EBF-Tropic | 2.25 | 1.21 | 0.36 | 1.03 | 0.26 | 0.98 | 0.24 | 0.58 | 0.08 | 0.70 | 0.12 |
| | DBF-MidL_T | 5.11 | 1.11 | 0.06 | 1.06 | 0.04 | 1.08 | 0.04 | 4.47 | 0.86 | 0.30 | 0.00 |
| | ENF-Bor_WT | 3.61 | 0.94 | 0.07 | 0.80 | 0.06 | 0.66 | 0.04 | 3.07 | 0.83 | 0.30 | 0.01 |
| | ENF-MidL_T | 3.50 | 1.16 | 0.10 | 1.00 | 0.07 | 0.74 | 0.05 | 3.02 | 0.77 | 0.36 | 0.01 |
| | ENF-Trans_E | 3.25 | 1.20 | 0.15 | 0.94 | 0.09 | 0.71 | 0.07 | 2.46 | 0.67 | 0.40 | 0.02 |
| | SAV-Trans_E | 2.05 | 0.57 | 0.07 | 0.54 | 0.09 | 0.52 | 0.07 | 1.71 | 0.75 | 0.30 | 0.02 |
| | CRO-MidL_T | 4.75 | 1.03 | 0.06 | 0.93 | 0.04 | 1.47 | 0.09 | 3.98 | 0.76 | 0.74 | 0.03 |
| NDVI | EBF-Tropic | 0.79 | | | 0.04 | 0.02 | 0.15 | 0.32 | 0.29 | 0.29 | 0.62 | 0.74 |
| | DBF-MidL_T | 1.89 | | | 0.33 | 0.07 | 0.80 | 0.38 | 0.92 | 0.56 | 0.08 | 0.01 |
| | ENF-Bor_WT | 1.08 | | | 0.09 | 0.01 | 0.37 | 0.25 | 0.74 | 0.70 | 0.05 | 0.00 |
| | ENF-MidL_T | 1.02 | | | 0.21 | 0.10 | 0.47 | 0.49 | 0.47 | 0.38 | 0.05 | 0.00 |
| | ENF-Trans_E | 0.64 | | | 0.19 | 0.16 | 0.37 | 0.38 | 0.23 | 0.46 | 0.03 | 0.01 |
| | SAV-Trans_E | 1.41 | | | 0.19 | 0.05 | 0.41 | 0.22 | 0.78 | 0.71 | 0.03 | 0.00 |
| | CRO-MidL_T | 1.42 | | | 0.24 | 0.06 | 0.65 | 0.36 | 0.76 | 0.58 | 0.04 | 0.00 |
| EVI | EBF-Tropic | 1.03 | | | 0.15 | 0.07 | 0.14 | 0.04 | 0.66 | 0.77 | 0.33 | 0.25 |
| | DBF-MidL_T | 3.59 | | | 0.48 | 0.04 | 1.38 | 0.33 | 2.08 | 0.63 | 0.07 | 0.00 |
| | ENF-Bor_WT | 1.44 | | | 0.08 | 0.00 | 0.45 | 0.24 | 0.86 | 0.74 | 0.05 | 0.00 |
| | ENF-MidL_T | 1.40 | | | 0.19 | 0.03 | 0.48 | 0.19 | 0.88 | 0.76 | 0.08 | 0.01 |
| | ENF-Trans_E | 0.53 | | | 0.15 | 0.09 | 0.22 | 0.32 | 0.23 | 0.59 | 0.13 | 0.07 |
| | SAV-Trans_E | 1.34 | | | 0.21 | 0.05 | 0.32 | 0.30 | 0.77 | 0.65 | 0.03 | 0.00 |
| | CRO-MidL_T | 2.76 | | | 0.42 | 0.03 | 1.15 | 0.21 | 1.63 | 0.75 | 0.11 | 0.00 |
| NIRv | EBF-Tropic | 1.35 | | | 0.19 | 0.07 | 0.17 | 0.05 | 0.84 | 0.77 | 0.43 | 0.22 |
| | DBF-MidL_T | 4.02 | | | 0.56 | 0.04 | 1.61 | 0.39 | 2.27 | 0.57 | 0.11 | 0.00 |
| | ENF-Bor_WT | 1.38 | | | 0.08 | 0.00 | 0.45 | 0.22 | 0.85 | 0.76 | 0.07 | 0.00 |
| | ENF-MidL_T | 1.36 | | | 0.17 | 0.03 | 0.50 | 0.23 | 0.89 | 0.72 | 0.07 | 0.01 |
| | ENF-Trans_E | 0.48 | | | 0.14 | 0.10 | 0.22 | 0.36 | 0.21 | 0.54 | 0.10 | 0.07 |
| | SAV-Trans_E | 1.36 | | | 0.21 | 0.05 | 0.31 | 0.32 | 0.75 | 0.63 | 0.04 | 0.00 |
| | CRO-MidL_T | 3.15 | | | 0.42 | 0.03 | 1.28 | 0.23 | 1.88 | 0.75 | 0.15 | 0.00 |
| NIRvP | EBF-Tropic | 2.06 | | | 0.42 | 0.15 | 0.42 | 0.15 | 1.13 | 0.63 | 0.54 | 0.14 |
| | DBF-MidL_T | 4.93 | | | 1.43 | 0.16 | 1.85 | 0.31 | 2.34 | 0.54 | 0.13 | 0.00 |
| | ENF-Bor_WT | 2.17 | | | 0.24 | 0.04 | 0.75 | 0.20 | 1.40 | 0.75 | 0.13 | 0.00 |
| | ENF-MidL_T | 2.38 | | | 0.71 | 0.22 | 0.82 | 0.29 | 1.15 | 0.46 | 0.04 | 0.00 |
| | ENF-Trans_E | 1.65 | | | 0.44 | 0.14 | 0.50 | 0.29 | 0.88 | 0.57 | 0.09 | 0.01 |
| | SAV-Trans_E | 1.49 | | | 0.43 | 0.18 | 0.49 | 0.30 | 0.86 | 0.52 | 0.01 | 0.00 |
| | CRO-MidL_T | 4.01 | | | 0.86 | 0.08 | 1.55 | 0.33 | 2.07 | 0.56 | 0.10 | 0.00 |
| SIF | EBF-Tropic | 1.47 | | | 0.53 | 0.22 | 0.47 | 0.17 | 0.89 | 0.61 | 0.11 | 0.01 |
| | DBF-MidL_T | 4.31 | | | 0.72 | 0.05 | 1.41 | 0.18 | 3.12 | 0.78 | 0.13 | 0.00 |
| | ENF-Bor_WT | 2.12 | | | 0.43 | 0.06 | 0.91 | 0.43 | 1.27 | 0.50 | 0.12 | 0.01 |
| | ENF-MidL_T | 2.55 | | | 0.63 | 0.09 | 0.84 | 0.17 | 1.71 | 0.73 | 0.09 | 0.00 |
| | ENF-Trans_E | 1.44 | | | 0.30 | 0.06 | 0.29 | 0.06 | 1.15 | 0.88 | 0.00 | 0.00 |
| | SAV-Trans_E | 1.68 | | | 0.26 | 0.03 | 0.48 | 0.13 | 1.48 | 0.84 | 0.03 | 0.00 |
| | CRO-MidL_T | 3.83 | | | 0.67 | 0.05 | 1.32 | 0.16 | 2.78 | 0.79 | 0.15 | 0.00 |





| | | All | Daily | | Weekly | | Monthly | | Annual | | Interannual | |
|---|---|---|---|---|---|---|---|---|---|---|---|---|
| Obs | EBF-Tropic | 2.25 | 1.21 | 0.36 | 1.03 | 0.26 | 0.98 | 0.24 | 0.58 | 0.08 | 0.70 | 0.12 |
| | DBF-MidL_T | 5.11 | 1.11 | 0.06 | 1.06 | 0.04 | 1.08 | 0.04 | 4.47 | 0.86 | 0.30 | 0.00 |
| | ENF-Bor_WT | 3.61 | 0.94 | 0.07 | 0.80 | 0.06 | 0.66 | 0.04 | 3.07 | 0.83 | 0.30 | 0.01 |
| | ENF-MidL_T | 3.50 | 1.16 | 0.10 | 1.00 | 0.07 | 0.74 | 0.05 | 3.02 | 0.77 | 0.36 | 0.01 |
| | ENF-Trans_E | 3.25 | 1.20 | 0.15 | 0.94 | 0.09 | 0.71 | 0.07 | 2.46 | 0.67 | 0.40 | 0.02 |
| | SAV-Trans_E | 2.05 | 0.57 | 0.07 | 0.54 | 0.09 | 0.52 | 0.07 | 1.71 | 0.75 | 0.30 | 0.02 |
| | CRO-MidL_T | 4.75 | 1.03 | 0.06 | 0.93 | 0.04 | 1.47 | 0.09 | 3.98 | 0.76 | 0.74 | 0.03 |
| FluxCom$_{RS}$ | EBF-Tropic | 0.49 | | | 0.19 | 0.20 | 0.12 | 0.08 | 0.37 | 0.72 | 0.03 | 0.00 |
| | DBF-MidL_T | 3.07 | | | 0.45 | 0.03 | 0.71 | 0.08 | 2.78 | 0.90 | 0.09 | 0.00 |
| | ENF-Bor_WT | 2.90 | | | 0.33 | 0.02 | 0.40 | 0.02 | 2.75 | 0.96 | 0.08 | 0.00 |
| | ENF-MidL_T | 2.95 | | | 0.45 | 0.03 | 0.53 | 0.04 | 2.70 | 0.93 | 0.10 | 0.00 |
| | ENF-Trans_E | 1.87 | | | 0.28 | 0.02 | 0.33 | 0.04 | 1.64 | 0.93 | 0.07 | 0.00 |
| | SAV-Trans_E | 0.92 | | | 0.18 | 0.03 | 0.34 | 0.12 | 0.79 | 0.86 | 0.02 | 0.00 |
| | CRO-MidL_T | 2.78 | | | 0.47 | 0.03 | 0.69 | 0.08 | 2.44 | 0.88 | 0.10 | 0.00 |
| FluxCom$_{RSMet}$ | EBF-Tropic | 0.75 | 0.53 | 0.59 | 0.22 | 0.10 | 0.22 | 0.10 | 0.31 | 0.21 | 0.05 | 0.01 |
| | DBF-MidL_T | 3.05 | 0.42 | 0.02 | 0.26 | 0.01 | 0.41 | 0.02 | 2.95 | 0.95 | 0.03 | 0.00 |
| | ENF-Bor_WT | 2.53 | 0.35 | 0.02 | 0.20 | 0.01 | 0.27 | 0.01 | 2.49 | 0.97 | 0.01 | 0.00 |
| | ENF-MidL_T | 2.88 | 0.42 | 0.02 | 0.24 | 0.01 | 0.29 | 0.01 | 2.82 | 0.96 | 0.05 | 0.00 |
| | ENF-Trans_E | 1.90 | 0.22 | 0.01 | 0.15 | 0.01 | 0.19 | 0.01 | 1.86 | 0.97 | 0.03 | 0.00 |
| | SAV-Trans_E | 1.07 | 0.20 | 0.04 | 0.14 | 0.02 | 0.20 | 0.04 | 1.00 | 0.91 | 0.06 | 0.00 |
| | CRO-MidL_T | 2.94 | 0.45 | 0.03 | 0.24 | 0.01 | 0.29 | 0.01 | 2.87 | 0.96 | 0.03 | 0.00 |
| MOD17 | EBF-Tropic | 3.29 | | | 1.64 | 0.39 | 1.26 | 0.23 | 1.61 | 0.38 | 0.07 | 0.00 |
| | DBF-MidL_T | 3.61 | | | 1.21 | 0.24 | 0.97 | 0.12 | 2.23 | 0.63 | 0.07 | 0.00 |
| | ENF-Bor_WT | 3.24 | | | 1.05 | 0.23 | 0.92 | 0.17 | 1.84 | 0.60 | 0.06 | 0.00 |
| | ENF-MidL_T | 2.89 | | | 0.98 | 0.21 | 0.79 | 0.13 | 1.77 | 0.64 | 0.04 | 0.00 |
| | ENF-Trans_E | 2.35 | | | 0.74 | 0.16 | 0.67 | 0.13 | 1.57 | 0.71 | 0.06 | 0.00 |
| | SAV-Trans_E | 1.81 | | | 0.47 | 0.09 | 0.50 | 0.15 | 1.33 | 0.74 | 0.03 | 0.00 |
| | CRO-MidL_T | 2.69 | | | 0.91 | 0.18 | 0.93 | 0.19 | 1.71 | 0.63 | 0.09 | 0.00 |
| LSA SAF | EBF-Tropic | 3.20 | 2.20 | 0.57 | 1.00 | 0.11 | 0.67 | 0.06 | 1.49 | 0.26 | 0.31 | 0.01 |
| | DBF-MidL_T | 6.72 | 2.47 | 0.15 | 1.46 | 0.06 | 0.87 | 0.03 | 5.72 | 0.77 | 0.10 | 0.00 |
| | ENF-Bor_WT | 5.45 | 2.07 | 0.15 | 1.27 | 0.06 | 0.94 | 0.03 | 4.59 | 0.76 | 0.14 | 0.00 |
| | ENF-MidL_T | 4.74 | 2.00 | 0.19 | 1.29 | 0.07 | 0.62 | 0.02 | 3.92 | 0.72 | 0.14 | 0.00 |
| | ENF-Trans_E | 2.66 | 1.03 | 0.18 | 0.58 | 0.05 | 0.33 | 0.02 | 2.26 | 0.75 | 0.27 | 0.01 |
| | SAV-Trans_E | 1.57 | 0.71 | 0.15 | 0.51 | 0.12 | 0.33 | 0.07 | 1.19 | 0.67 | 0.13 | 0.01 |
| | CRO-MidL_T | 3.06 | 1.21 | 0.16 | 0.72 | 0.05 | 0.68 | 0.05 | 2.52 | 0.75 | 0.15 | 0.00 |
| ISBA | EBF-Tropic | 2.94 | 1.95 | 0.53 | 0.94 | 0.12 | 0.57 | 0.05 | 1.49 | 0.30 | 0.07 | 0.00 |
| | DBF-MidL_T | 4.47 | 1.37 | 0.10 | 0.95 | 0.05 | 0.88 | 0.04 | 3.88 | 0.80 | 0.49 | 0.01 |
| | ENF-Bor_WT | 2.76 | 0.79 | 0.08 | 0.60 | 0.05 | 0.43 | 0.02 | 2.43 | 0.84 | 0.15 | 0.00 |
| | ENF-MidL_T | 2.89 | 0.94 | 0.11 | 0.70 | 0.06 | 0.39 | 0.02 | 2.49 | 0.81 | 0.14 | 0.00 |
| | ENF-Trans_E | 3.72 | 1.17 | 0.10 | 0.72 | 0.05 | 0.47 | 0.03 | 3.22 | 0.84 | 0.25 | 0.01 |
| | SAV-Trans_E | 2.34 | 0.40 | 0.06 | 0.40 | 0.08 | 0.53 | 0.05 | 2.01 | 0.77 | 0.40 | 0.03 |
| | CRO-MidL_T | 4.26 | 1.42 | 0.13 | 0.95 | 0.07 | 1.14 | 0.08 | 3.34 | 0.72 | 0.48 | 0.01 |
| ORCHIDEE | EBF-Tropic | 1.32 | 0.89 | 0.55 | 0.33 | 0.08 | 0.45 | 0.15 | 0.56 | 0.21 | 0.04 | 0.00 |
| | DBF-MidL_T | 5.03 | 1.13 | 0.05 | 1.04 | 0.05 | 1.34 | 0.07 | 4.52 | 0.81 | 0.18 | 0.00 |
| | ENF-Bor_WT | 3.22 | 0.64 | 0.04 | 0.49 | 0.02 | 0.34 | 0.01 | 3.07 | 0.92 | 0.12 | 0.00 |
| | ENF-MidL_T | 2.47 | 0.63 | 0.07 | 0.51 | 0.04 | 0.34 | 0.02 | 2.25 | 0.86 | 0.13 | 0.00 |
| | ENF-Trans_E | 2.23 | 0.66 | 0.09 | 0.47 | 0.04 | 0.73 | 0.10 | 1.91 | 0.72 | 0.14 | 0.00 |
| | SAV-Trans_E | 2.41 | 0.53 | 0.05 | 0.54 | 0.05 | 0.55 | 0.06 | 2.19 | 0.83 | 0.61 | 0.05 |
| | CRO-MidL_T | 4.57 | 0.96 | 0.05 | 0.79 | 0.03 | 1.35 | 0.08 | 4.14 | 0.85 | 0.33 | 0.01 |

**Table 2.** Standard deviation of the observed and simulated GPP ($\mathrm{gC/m^2/d}$), decomposed in the different timescale components using SSA. The median values for 6 land cover types are reported here. The fraction of the total variability is given in the grey columns.



## GPP Covariance

| | Spatial Spatial | Seasonal Seasonal | Anomalies Anomalies | Spatial Seasonal | Spatial Anomalies | Seasonal Anomalies |
|---|---|---|---|---|---|---|
| Observation | 3.13 | 10.84 | 4.22 | 0.12 | -0.12 | -0.34 |
| NDVI | 2.13 | 1.77 | 0.55 | 0.04 | -0.04 | -0.01 |
| EVI | 2.87 | 5.07 | 0.81 | 0.00 | 0.00 | -0.04 |
| NIRv | 3.19 | 5.77 | 0.94 | -0.01 | 0.01 | -0.05 |
| NIRvP | 1.36 | 7.38 | 3.15 | 0.02 | -0.02 | -0.37 |
| SIF | 2.74 | 7.70 | 1.11 | -0.03 | 0.03 | 0.05 |
| FluxCom$_{RS}$ | 1.26 | 6.24 | 0.44 | 0.00 | 0.00 | -0.01 |
| FluxCom$_{RSMet}$ | 1.31 | 6.70 | 0.29 | 0.04 | -0.04 | -0.16 |
| MOD17 | 1.94 | 5.86 | 2.27 | 0.03 | -0.03 | -0.15 |
| LSA SAF | 5.01 | 13.53 | 5.66 | 0.17 | -0.16 | -0.44 |
| ISBA | 2.13 | 8.31 | 3.43 | 0.09 | -0.09 | -0.31 |
| ORCHIDEE | 1.81 | 10.09 | 3.08 | 0.16 | -0.16 | -0.71 |

**Table 3.** (Co-)variance of the observed and simulated GPP (gC/m$^2$/d), decomposed in the inter-site, seasonal and anomalies (obtained after subtracting the spatial and seasonal component) components. Analysis of the concatenated timeseries.

| | Daily | | | | | Weekly | | | | | Monthly | | | | |
|---|---|---|---|---|---|---|---|---|---|---|---|---|---|---|---|
| | D | W | M | S | I | D | W | M | S | I | D | W | M | S | I |
| Observation | 0.96 | 0.11 | 0.00 | 0.02 | 0.00 | 0.11 | 0.62 | 0.07 | 0.06 | 0.00 | 0.00 | 0.07 | 0.56 | 0.12 | 0.00 |
| NDVI | 0.00 | 0.00 | 0.00 | 0.00 | 0.00 | 0.00 | 0.05 | 0.06 | 0.07 | 0.00 | 0.00 | 0.06 | 0.23 | 0.22 | 0.01 |
| EVI | 0.00 | 0.00 | 0.00 | 0.00 | 0.00 | 0.00 | 0.04 | 0.05 | 0.11 | 0.00 | 0.00 | 0.05 | 0.28 | 0.42 | 0.02 |
| NIRv | 0.00 | 0.00 | 0.00 | 0.00 | 0.00 | 0.00 | 0.05 | 0.06 | 0.12 | 0.00 | 0.00 | 0.06 | 0.29 | 0.41 | 0.02 |
| NIRvP | 0.00 | 0.00 | 0.00 | 0.00 | 0.00 | 0.00 | 0.37 | 0.33 | 0.38 | 0.01 | 0.00 | 0.33 | 0.67 | 0.67 | 0.01 |
| SIF | 0.00 | 0.00 | 0.00 | 0.00 | 0.00 | 0.00 | 0.32 | 0.14 | 0.21 | 0.01 | 0.00 | 0.14 | 0.81 | 0.74 | 0.02 |
| FluxCom$_{RS}$ | 0.00 | 0.00 | 0.00 | 0.00 | 0.00 | 0.00 | 0.12 | 0.04 | 0.03 | 0.00 | 0.00 | 0.04 | 0.21 | 0.21 | 0.01 |
| FluxCom$_{RSMet}$ | 0.12 | 0.01 | 0.00 | 0.00 | 0.00 | 0.01 | 0.05 | 0.00 | 0.00 | 0.00 | 0.00 | 0.00 | 0.06 | 0.01 | 0.00 |
| MOD17 | 0.00 | 0.00 | 0.00 | 0.00 | 0.00 | 0.00 | 0.88 | 0.41 | 0.49 | 0.01 | 0.00 | 0.41 | 0.60 | 0.51 | 0.01 |
| LSA SAF | 2.01 | 0.24 | 0.03 | 0.00 | 0.00 | 0.24 | 0.65 | 0.07 | 0.01 | 0.00 | 0.03 | 0.07 | 0.38 | 0.02 | 0.00 |
| ISBA | 0.86 | 0.11 | 0.01 | 0.00 | 0.00 | 0.11 | 0.47 | 0.06 | 0.00 | 0.00 | 0.01 | 0.06 | 0.21 | 0.03 | 0.01 |
| ORCHIDEE | 0.53 | 0.06 | 0.00 | 0.00 | 0.00 | 0.06 | 0.34 | 0.04 | 0.00 | 0.00 | 0.00 | 0.04 | 0.48 | 0.04 | 0.00 |

| | Seasonal | | | | | Interannual | | | | |
|---|---|---|---|---|---|---|---|---|---|---|
| | D | W | M | S | I | D | W | M | S | I |
| Observation | 0.02 | 0.06 | 0.12 | 8.31 | 0.03 | 0.00 | 0.00 | 0.00 | 0.03 | 0.08 |
| NDVI | 0.00 | 0.07 | 0.22 | 0.43 | 0.02 | 0.00 | 0.00 | 0.01 | 0.02 | 0.00 |
| EVI | 0.00 | 0.11 | 0.42 | 1.22 | 0.03 | 0.00 | 0.00 | 0.02 | 0.03 | 0.00 |
| NIRv | 0.00 | 0.12 | 0.41 | 1.08 | 0.03 | 0.00 | 0.00 | 0.02 | 0.03 | 0.01 |
| NIRvP | 0.00 | 0.38 | 0.67 | 1.79 | 0.03 | 0.00 | 0.01 | 0.01 | 0.03 | 0.00 |
| SIF | 0.00 | 0.21 | 0.74 | 2.93 | 0.05 | 0.00 | 0.01 | 0.02 | 0.05 | 0.01 |
| FluxCom$_{RS}$ | 0.00 | 0.03 | 0.21 | 5.03 | 0.07 | 0.00 | 0.00 | 0.01 | 0.07 | 0.01 |
| FluxCom$_{RSMet}$ | 0.00 | 0.00 | 0.01 | 6.38 | 0.00 | 0.00 | 0.00 | 0.00 | 0.00 | 0.00 |
| MOD17 | 0.00 | 0.49 | 0.51 | 2.63 | 0.03 | 0.00 | 0.01 | 0.01 | 0.03 | 0.00 |
| LSA SAF | 0.00 | 0.01 | 0.02 | 8.66 | 0.01 | 0.00 | 0.00 | 0.00 | 0.01 | 0.02 |
| ISBA | 0.00 | 0.00 | 0.03 | 6.74 | 0.02 | 0.00 | 0.00 | 0.01 | 0.02 | 0.04 |
| ORCHIDEE | 0.00 | 0.00 | 0.04 | 7.43 | 0.01 | 0.00 | 0.00 | 0.00 | 0.01 | 0.02 |

**Table 4.** (Co-)variance of the observed and simulated GPP (gC/m$^2$/d), of the daily, weekly, monthly, seasonal and inter-annual timescale (obtained with SSA). The median for all sites is given here.



**Covariance of GPP Drivers**

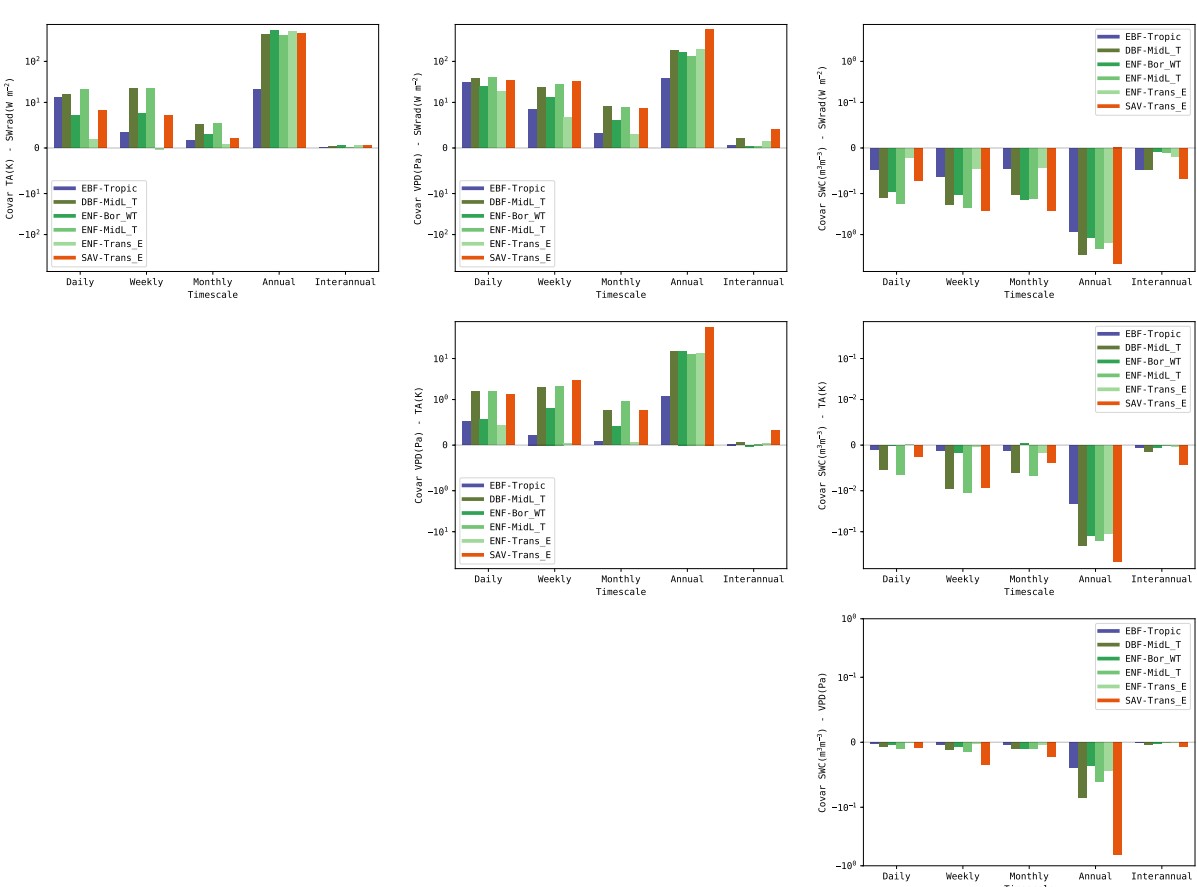

**Figure A4.** Covariance of the GPP drivers: SWrad, TA, VPD and SWC. Median of the sites, classified per land cover type.





## Correlation between GPP and its drivers

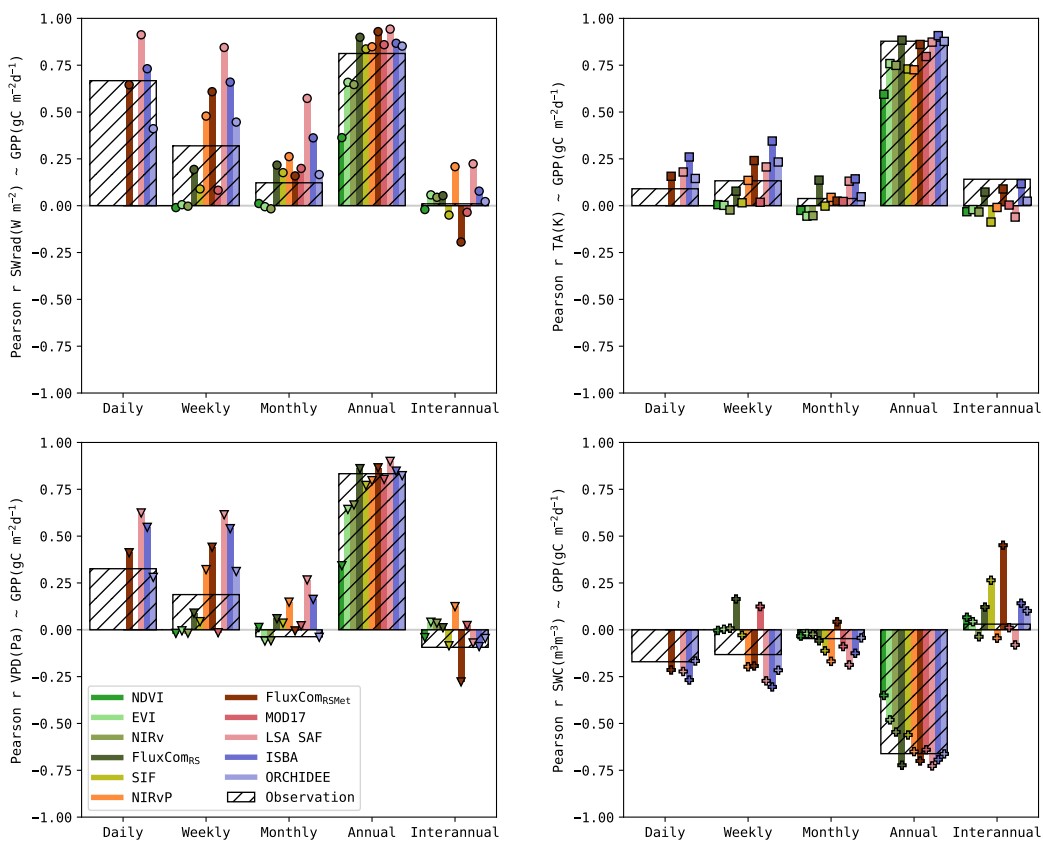

**Figure A5.** Correlation of the simulated GPP and its drivers (median for all sites). Correlation based on observations are in the hashed bars. The colored barplots indicate the correlation in the simulations.



**Figure A6.** Pearson r of the simulated GPP and its drivers at weekly, monthly and seasonal scale. Correlation based on observations are in the hashed bars and gray bars highlight the deviation for a land cover from the overall average. The colored barplots indicate the correlation in the simulations.



600 **Accuracy of the GPP-driver covariance**

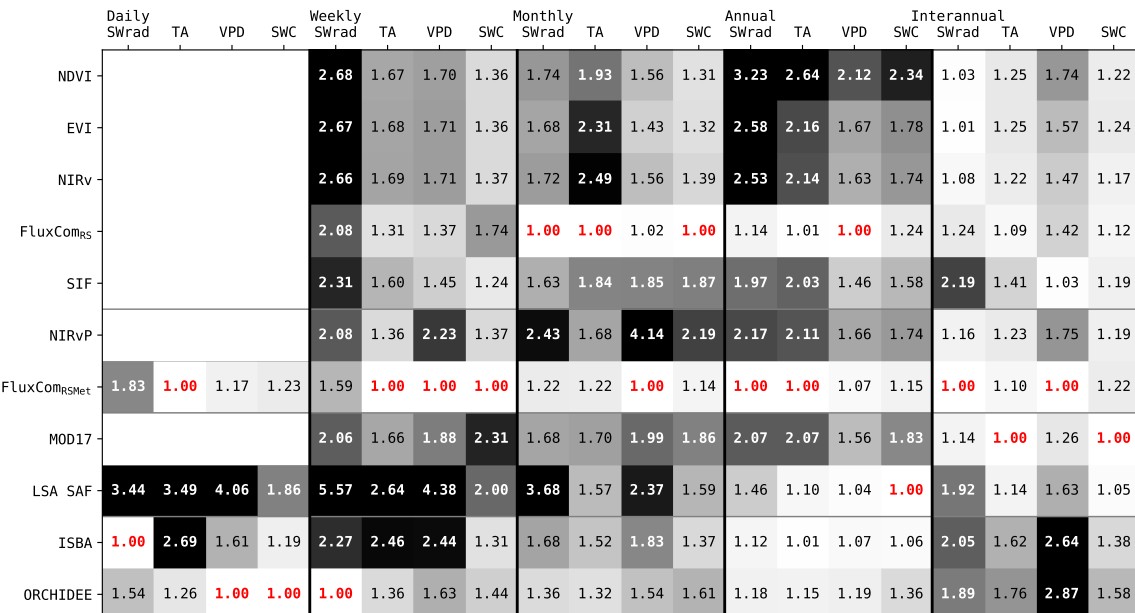

**Figure A7.** RMSE of the driver-GPP covariances at weekly, monthly and annual timescale. The values are relative to the lowest RMSE score
(best model has unitless value 1)







**Figure A8.** RMSE of the driver-GPP covariances at annual timescale, per biome. The values are relative to the lowest RMSE score (best model has unitless value 1)



**Phenology**

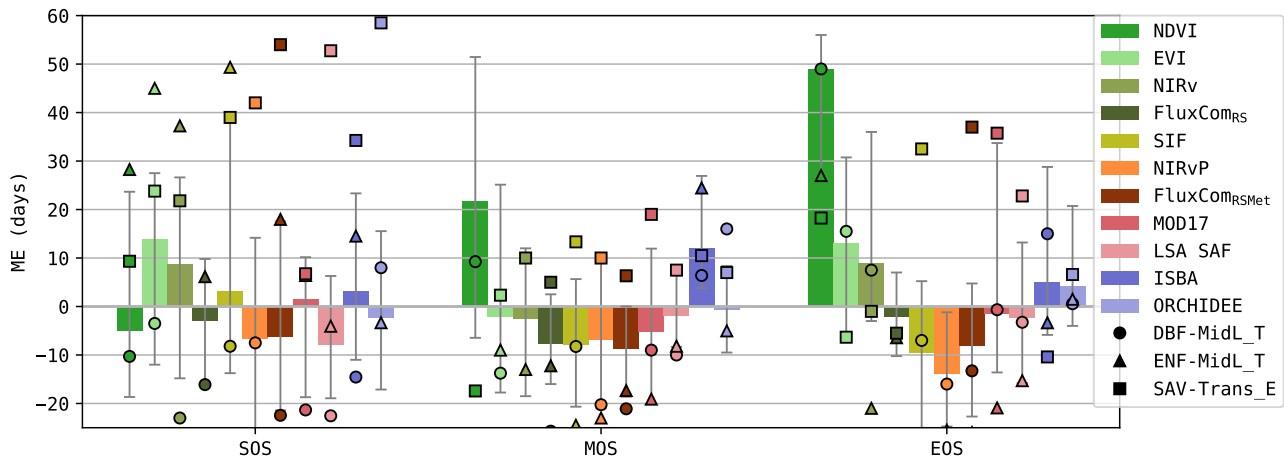

**Figure A9.** Mean errors (per site) in the timing of the start, max and end of the seasonal GPP cycle (SOS, MOS and EOS)



**Mean annual cycle**

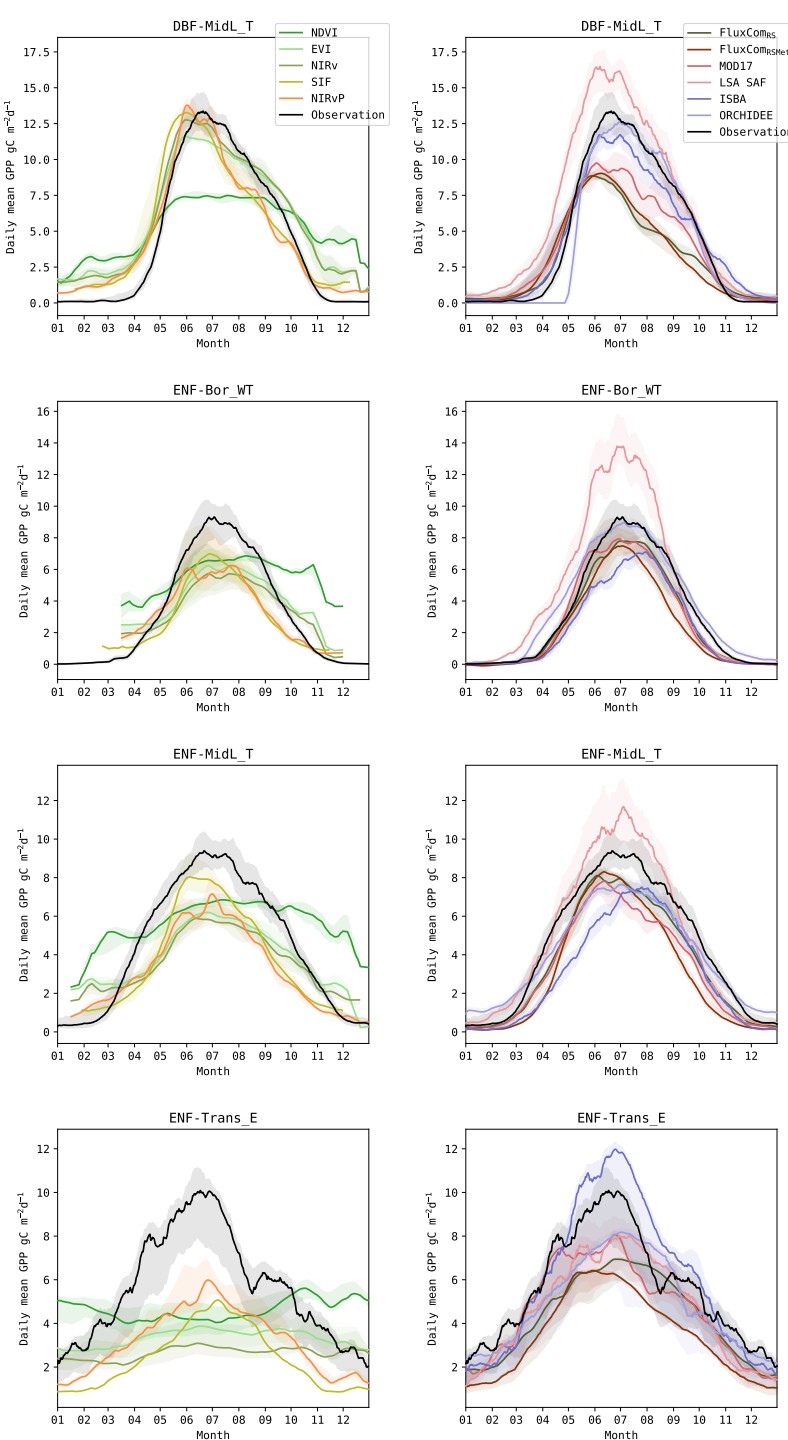

**Figure A10.** Annual GPP cycle in observations and models, for sites in the DBF-Midl_T, ENF-Bor_WT , ENF-Midl_T and ENF-Trans_E biomes. The lines show the median cycle, and the shaded area shows the 25-75 percentile. Timeseries of sites located at the southern hemisphere were shifted by 6 months, to match with the annual cycle of sites in the northern hemisphere.




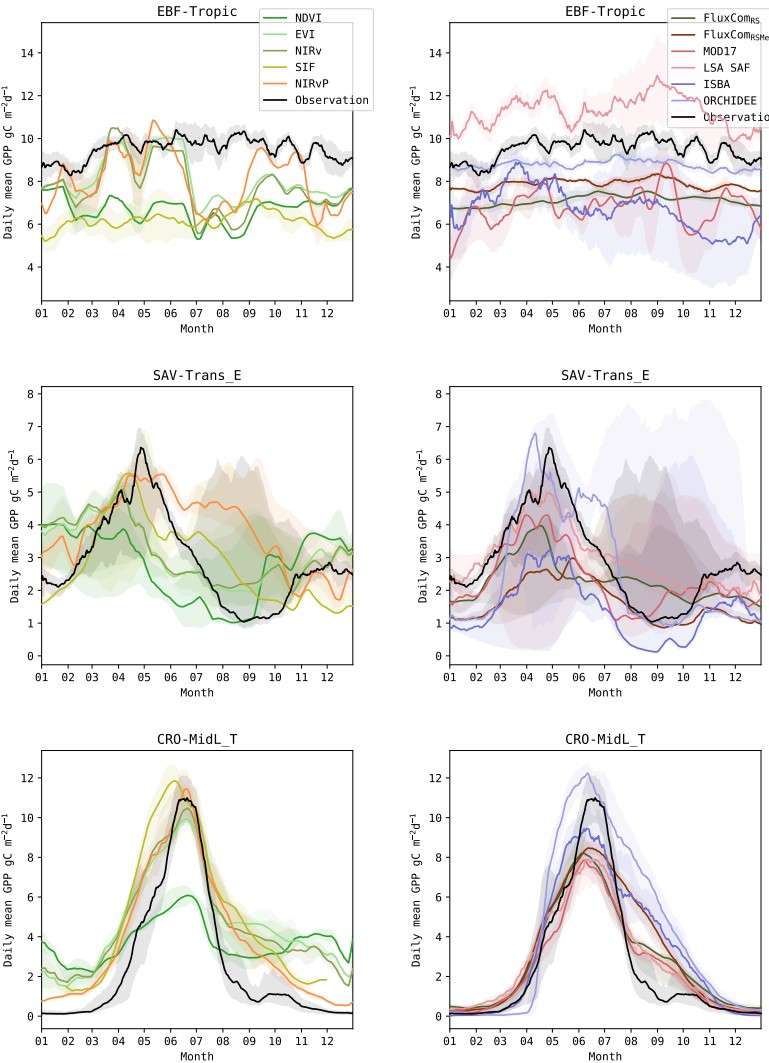

**Figure A11.** Annual GPP cycle in observations and models, for sites in the EBF-Tropic, SAV-Trans_E , and CRO-Midl_T biomes. The lines show the median cycle, and the shaded area shows the 25-75 percentile. Timeseries of sites located at the southern hemisphere were shifted by 6 months, to match with the annual cycle of sites in the northern hemisphere.



*Author contributions.* JDP: Conceptualization, Investigation, Analysis, Writing - original draft preparation; SW, AB: Investigation, Analysis, Writing - review & editing;JMB, LL, PC, AA, RH, MM: Writing - review & editing; FM, FGM, IJ, MB: Supervision, Project administration, Writing - review & editing

605

*Competing interests.* The authors declare to have no competing interests

*Acknowledgements.* The research presented in this paper is funded by BELSPO (Belgian Science Policy Office) in the frame of the STEREO III programme – projects ECOPROPHET (SR/00/334) and ECOPROPHECIES (SR/34/211). We thank the countless contributors behind the scenes of the FLUXNET2015 dataset (Pastorello et al., 2020) and the ICOS '2018 drought initiative' dataset (Drought 2018 Team and ICOS Ecosystem Thematic Centre, 2019). These publicly available datasets are the keystone of this study.

610



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
