# Peer review of "Temporal variability of observed and simulated gross primary productivity, modulated by vegetation state and hydrometeorological drivers"

_EGUsphere, 2023_

## Author Comment (AC1)

**Rebuttal BG-2023-994**
**Temporal variability of observed and simulated gross primary productivity, modulated by vegetation state and hydrometeorological drivers**

Jan De Pue, Sebastian Wieneke, Ana Bastos, José Miguel Barrios,

Liyang Liu, Philippe Ciais, Alirio Arboleda, Rafiq Hamdi, Maral Maleki,

Fabienne Maignan, Françoise Gellens-Meulenberghs, Ivan Janssens and Manuela Balzarolo

Handling Associate Editor: Paul Stoy

October 3, 2023
* * *
**Anonymous referee 1**

The manuscript aims to verify the variance exhibited by common GPP models and products in repsonse to the driving factors of GPP. The manuscript is well written in general. In my personal opinion, the abstract and discussion sections are excellent as they nicely summarize the key results in a crisp manner. I have few queries/suggestions for the consideration of the authors.

**Comment 1.1** — What is the reason for not including the accuracy assessment of the models? Is it because of the lack of independent data for validation? Idea about absolute accuracy would further help the readers to know about the GPP models.

> Indeed, we choose to avoid accuracy assessment, as there is a lack of independent data for validation. Though the dataset could be split for training and validation of the RS-based regression models, it remains unclear to what extend the dataset has been used to train/calibrate other models (e.g. FluxCom, MOD17, ISBA, ORCHIDEE, ...). To minimize the impact of this, we do not to focus on the accuracy, but on the variability instead.
>
> However, as the accuracy and bias could be of interest for the readers, we add those results to the supplement material:
>
> > " Note that the training data used here was also used in the evaluation of the model performance. Furthermore, most models in this study were directly or indirectly trained with data from eddy covariance towers (e.g., FluxCom (Jung et al., 2020), ORCHIDEE (Friend et al., 2007), ...). Consequently, it was not possible to ensure an independent validation of the models. To minimize impact on the study results, the evaluation was largely based on metrics that are not impacted by the slope of the linear regression (correlation and phenology analysis, see further below). Absolute errors and bias of the models were not evaluated in this study, as these indices are significantly affected by the overlap between training and evaluation data, but they

*are shown in the supplement material for completeness (Fig. R-1).*
*"*

[Figure]

[Figure]

Figure R-1: ME (left) and RMSE (right) of the GPP obtained by the models, validated with the in-situ observations. Bars show the median score accross the sites, errorbars indicate the 25 and 75 percentile. Note: this is not an independent validation.

**Comment 1.2** — Why the downscaled SIF product was considered in this study? It will have the effect of LUE model used in the downscaling and any artefacts of downscaling will be present. Spatial resolution is an issue especially when the SIF based GPP has to be compared with flux tower observations. But the downscaling procedure might have affected the spatiotemporal characteristics of the SIF product affecting the variance metrics studied in this work. If the effect of SIF to be studied, it is better to use the original datasets without much modification.

> We agree with the reviewer that the downscaled product contains artefacts of the downscaling methods, however as the reviewer addresses we decided to use the downscaled product because of the large Footprint of GOME-2 SIF and the fact that most of the sites used in this study are too heterogeous on a scale of 40 x 40 km. During the development of the methodology, for information on sun- induced fluorescence (SIF) we tested also OCO-2 data but due to its sparse global coverage (only a dozen of observations for all EC-Tower locations were available per year, resulting in a too low sample size) we decided to refrain from using this data. Also, SIF retrievals of TROPOMI were not used in this analysis due to its limited temporal coverage (2018-2019).
>
> In response to this remark, we address this in the materials & methods section:
>
> > *" For remotely sensed SIF data, we relied on the downscaled GOME-2 SIF product by Duveiller et al. (2020), given the coarse spatial resolution of the GOME-2 SIF product (>40 km), sparse global coveraged (only a dozen of GOME-2 observations for all tower locations were available per year) and the limited available timeseries of TROPOMI (starting in May 2018). The downscaling procedure involves a LUE methodology, involving NIRv, normalised difference water index (NDWI; Gao, 1996) and land surface temperature (LST) data from MODIS.*
> > *"*
>
> The artefacts of the downscaled product might have an influence on the results. We therefore stipulate again that we used the downscaled product and not the original GOME-2 SIF. However, we also point out that Duveiller et al. (2020) found a

high spatiotemporal agreement between the downscaled product and TROPOMI observations, supporting the choice to use this downscaled product:

> " Duveiller et al. (2020) demonstrated that this product has a high spatio-temporal agreement with TROPOMI SIF observations, so the impact of the artefacts due to the downscaling procedure are assumed to be limited. "

The uncertainties due to the downscaling is also acknowledged in the discussion:

> " It remains unclear in what measure the downscaling processing is responsible for the moderate SIF scores. Future missions with high resolution SIF, such as European Spatial Agency's Earth Explorer - FLEX (FLuorescence EXplorer, due to be launched in 2025) will provide further insights (Duveiller et al., 2020). "

**Comment 1.3** — What is the temporal frequency of the reflectance data (SPV) that were used to derive vegetation indices? I understand it to be 16-days. If yes, then it will strongly impact the variances at daily and monthly scales. Why can't daily/weekly data be used?

Indeed this needs clarification. The following explanation was added to the text:

> " Remote sensing data was gathered from SPOT Vegetation+PROBA-V (SPV) for each tower location (the nearest pixel). This data product has a 10-day interval and 1km resolution. The SPV decadal synthesis product is derived using the Maximum Value Composite after quality check of SPV native data and give the best reflectance value on the 10-day time. Though daily data is available, it was not used here. The use of daily data would introduce gaps and noise in the SPV time series (in case of cloudy conditions at satellite overpass time, for instance) while not adding significant information on the vegetation status throughout the study period. "

Additionally, we ensured to specify the temporal and spatial resolution of all remote sensing products in the manuscript.

**Comment 1.4** — As you rightly pointed out, empirical models (Vegetation Indices models) must be developed for each landcover/climate zone combination. There can be significant improvement in the results had it been the case rather than having one generalized regression model. Landcover specific models can be developed to see if there are any changes in the performance of the VI-based models.

In response to this comment, a limited evaluation of PFT-specific regression models was performed. The sites were grouped per PFT class, and the same regression methodology as described in the paper was used to derive PFT-specific models. In the text, we added:

> " Results from explorative tests with PFT-specific regression models are shown in the supplement materials (Fig. R-2 and Fig. R-3). They indicated that improved results were largely caused by improved spatial correlation. The variability of the seasonal signal and anomalies remained underestimated. "

The following supplement material was added:

[Figure]

Figure R-2: Quantile regression models to estimate GPP from NDVI. The global regression (in black) and PFT-specific regressions (in colors) are shown. The same was performed for EVI, NIRv, SIF and NIRvP.

[Figure]

Figure R-3: Taylor diagram of the simulated GPP with global and PFT-specific regression. The validation of the full dataset is shown (top left), as well as the spatial component (top right), seasonal component (bottom left) and anomalies (bottom right)

**Comment 1.5** — In Section 2.2, it will be beneficial to add a brief summary of the tests conducted to test the validity of ERA-5 data in place of in-situ observations. If not in the main article, include it as appendix.

To demonstrate the validity of replacing in-situ meteorological forcing variables with ERA-5 data, the following material was added to the supplement:

> " *Given the gaps in the meteorological observations at the sites, used to force the models, ERA5 data was used to replace some of these variables: air temperature, atmospheric humidity, wind speed, and atmospheric pressure. Simulations with ISBA were performed using this configuration, and compared with simulations using only the tower observations (for sites that had $< 5\%$ gaps in the timeseries), to verify that the impact on the simulated GPP was limited. An example of simulations with both configurations is given in Fig. R-4, validation indices of both simulations compared to the observed GPP is given in Tab. R-1, and metrics of the similarity between both simulations are given in Tab. R-2. From these results, we conclude that the impact of using ERA5 for some of the forcing variables is negligible.* "

[Figure]

Figure R-4: Timeseries of the observed GPP at CZ-RAJ, compared to ISBA simulations with ERA5 variables (as used in the study), and ISBA simulations forced by the in situ meteorological observations (ISBA_Tower). The two simulations are largely overlapping.

| | | ME (gC/m$^2$/d) | RMSE (gC/m$^2$/d) | r (-) | $r_{seas}$ (-) | $r_{anom}$ (-) |
|---|---|---|---|---|---|---|
| CZ-RAJ | ISBA | -1.01 | 2.24 | 0.82 | 0.94 | 0.45 |
| | ISBA_Tower | -1.28 | 2.48 | 0.80 | 0.93 | 0.40 |
| DE-RuW | ISBA | -2.71 | 3.50 | 0.78 | 0.95 | 0.45 |
| | ISBA_Tower | -2.67 | 3.48 | 0.77 | 0.95 | 0.46 |
| FR-EM2 | ISBA | -2.01 | 5.98 | 0.54 | 0.92 | -0.14 |
| | ISBA_Tower | -1.68 | 5.96 | 0.53 | 0.91 | -0.17 |
| FR-Hes | ISBA | 0.36 | 2.88 | 0.85 | 0.94 | 0.54 |
| | ISBA_Tower | 0.52 | 2.75 | 0.86 | 0.96 | 0.55 |
| DE-Geb | ISBA | 0.81 | 3.70 | 0.66 | 0.85 | 0.33 |
| | ISBA_Tower | 0.68 | 3.59 | 0.66 | 0.87 | 0.34 |
| IT-SR2 | ISBA | 0.60 | 2.07 | 0.85 | 0.98 | 0.56 |
| | ISBA_Tower | 0.52 | 2.03 | 0.85 | 0.99 | 0.57 |
| CZ-Stn | ISBA | 0.11 | 2.53 | 0.87 | 0.98 | 0.49 |
| | ISBA_Tower | 0.16 | 2.49 | 0.87 | 0.98 | 0.51 |
| DE-Hte | ISBA | -0.44 | 2.28 | 0.62 | 0.90 | 0.04 |
| | ISBA_Tower | -0.34 | 2.31 | 0.62 | 0.90 | 0.05 |
| ES-LM1 | ISBA | -0.72 | 1.39 | 0.87 | 0.99 | 0.69 |
| | ISBA_Tower | -0.78 | 1.45 | 0.87 | 0.98 | 0.69 |
| ES-LM2 | ISBA | -0.75 | 1.38 | 0.87 | 0.98 | 0.67 |
| | ISBA_Tower | -0.80 | 1.46 | 0.86 | 0.95 | 0.67 |

Table R-1: Validation of GPP similated with ISBA (partly with ERA5 forcing variable) and ISBA_Tower (with tower forcing). The 10 sites were selected based on the good quality of the meteorological observations.

|        | ME | RMSE | r | $r_{seas}$ | $r_{anom}$ |
|--------|-----------|-----------|-------|--------|--------|
|        | (gC/m$^2$/d) | (gC/m$^2$/d) | (-) | (-) | (-) |
| CZ-RAJ | -0.266 | 0.377 | 0.996 | 0.999 | 0.988 |
| DE-RuW | 0.038 | 0.158 | 0.998 | 1.000 | 0.995 |
| FR-EM2 | 0.332 | 0.728 | 0.992 | 0.996 | 0.980 |
| FR-Hes | 0.161 | 0.504 | 0.995 | 0.996 | 0.988 |
| DE-Geb | -0.136 | 0.487 | 0.995 | 0.999 | 0.988 |
| IT-SR2 | -0.083 | 0.500 | 0.991 | 0.993 | 0.985 |
| CZ-Stn | 0.059 | 0.661 | 0.990 | 0.996 | 0.973 |
| DE-Hte | 0.088 | 0.322 | 0.993 | 0.999 | 0.980 |
| ES-LM1 | -0.066 | 0.368 | 0.989 | 0.993 | 0.980 |
| ES-LM2 | -0.056 | 0.351 | 0.989 | 0.993 | 0.979 |

Table R-2: Metrics to quantify the similarity between GPP similated with ISBA (partly with ERA5 forcing variable) and ISBA_Tower (with tower forcing). The 10 sites were selected based on the good quality of the meteorological observations.

**Comment 1.6** — In page 6, Equation (4), was PAR considered as a constant fraction of incoming solar radiation? A brief mention about PAR estimation will be beneficial.

Indeed, PAR was considered as a constant fraction of SWrad. This is now clarified in the text:

> "... where PAR is the daily mean photosynthetically active radiation, calculated as a constant fraction (0.45) of the in-situ incoming shortwave radiation observations (Howell et al., 1983)."

**Comment 1.7** — Equation 6 can be explained with an example.

The following example was added to clarify Eq.6:

[Figure]

Figure R-5: Illustration of the GPP data (top subplot) decomposition into inter-site (i.e., spatial) component (second subplot), seasonal component (third subplot) and the component associated with the anomalies (bottom subplot). This example shows the observed GPP from DE-Spw, RU-Fyo and US-SRM (left to right).

**Comment 1.8** — When referring to the appendix in the main text, always refer the figure or table to be looked upon. It became little difficult to identify which figure/table

to refer. Further, it was difficult to read the figures given in the appendix. Please improve the readability of all the figures.

This was revised, we refer to the figures and tables, instead of the section in the supplement.

**References**

G. Duveiller, F. Filipponi, S. Walther, P. Köhler, C. Frankenberg, L. Guanter, and A. Cescatti. A spatially downscaled sun-induced fluorescence global product for enhanced monitoring of vegetation productivity. *Earth System Science Data*, 12 (2):1101–1116, 2020.

A. D. Friend, A. Arneth, N. Y. Kiang, M. Lomas, J. Ogee, C. Rödenbeck, S. W. Running, J.-D. SANTAREN, S. Sitch, N. Viovy, et al. Fluxnet and modelling the global carbon cycle. *Global Change Biology*, 13(3):610–633, 2007.

B.-C. Gao. Ndwi—a normalized difference water index for remote sensing of vegetation liquid water from space. *Remote sensing of environment*, 58(3):257–266, 1996.

T. Howell, D. Meek, and J. Hatfield. Relationship of photosynthetically active radiation to shortwave radiation in the san joaquin valley. *Agricultural Meteorology*, 28(2):157–175, 1983.

M. Jung, C. Schwalm, M. Migliavacca, S. Walther, G. Camps-Valls, S. Koirala, P. Anthoni, S. Besnard, P. Bodesheim, N. Carvalhais, et al. Scaling carbon fluxes from eddy covariance sites to globe: synthesis and evaluation of the fluxcom approach. *Biogeosciences*, 17(5):1343–1365, 2020.

C. Papagiannopoulou, D. G. Miralles, M. Demuzere, N. E. Verhoest, and W. Waegeman. Global hydro-climatic biomes identified via multitask learning. *Geoscientific Model Development*, 11(10):4139–4153, 2018.

---

## Author Comment (AC2)

**Rebuttal BG-2023-994**
**Temporal variability of observed and simulated gross primary productivity, modulated by vegetation state and hydrometeorological drivers**

Jan De Pue, Sebastian Wieneke, Ana Bastos, José Miguel Barrios,

Liyang Liu, Philippe Ciais, Alirio Arboleda, Rafiq Hamdi, Maral Maleki,

Fabienne Maignan, Françoise Gellens-Meulenberghs, Ivan Janssens and Manuela Balzarolo

Handling Associate Editor: Paul Stoy

October 3, 2023
* * *
**Anonymous referee 2**

De Pue and others investigate multiple methods for estimating GPP against observations. The results are insightful and well written and I recommend the manuscript be published after considering the following rather minor comments.

**Comment 2.1** — How is 'homogeneous' defined in section 2.1?

This was further clarified in the text as follows:

> " *It was ensured that the sites had a homogeneous land cover, which could be captured by the remote sensing products. A site was considered homogeneous when in 1 x 1 km area surrounding the station location was dominated by a unique vegetation type (i.e., grassland, deciduous forest, evergreen forest). The site homogeneity was visually evaluated using high-resolution satellite images in Google Earth.* "

**Comment 2.2** — Explaining the Papagiannopoulou et al. (2018) delineations would be helpful because these are not in common usage. I see now that they are defined in the text; pointing to this text would help.

It is unclear which actions need to be taken to improve the text?

**Comment 2.3** — Please use the multiplication sign instead of the star for formal equations

Indeed, this was corrected. For example:

$$\text{EVI} = 2.5 \frac{R_{770-800} - R_{630-670}}{R_{770-800} + 6 \cdot R_{630-670} + 7.5 \cdot R_{460-475} + 1} \tag{R-1}$$

**Comment 2.4** — table 2: 'shortwave' can be used

Indeed, this was corrected.

**Comment 2.5** — the y axis in figure 3 is a bit confusing to me regarding the '-' symbols (and fig. 4, and 5)

The '-' symbol is used to indicate that the values on this axis are dimensionless. An example is the Pearson correlation. In principle, the unit could be given: $\mathrm{gC\,m^{-2}\,d^{-1}\,g^{-1}C\,m^2\,d}$. However, this is uncommon and unhelpful for the readability of the figures. The labels of the y-axis in these figures was kept as is.

**Comment 2.6** — Fig. 7 is a bit much to look at and I wonder if this analysis would be better off in an appendix.

This figure shows the different responses to hydrometeorological drivers in various biomes. It is discussed extensively in the manuscript. Though we agree that this is a dense figure, it is our opinion that it is insightful to have it in the main manuscript. We prefer not to move this figure to the supplements.

**References**

G. Duveiller, F. Filipponi, S. Walther, P. Köhler, C. Frankenberg, L. Guanter, and A. Cescatti. A spatially downscaled sun-induced fluorescence global product for enhanced monitoring of vegetation productivity. *Earth System Science Data*, 12 (2):1101–1116, 2020.

A. D. Friend, A. Arneth, N. Y. Kiang, M. Lomas, J. Ogee, C. Rödenbeck, S. W. Running, J.-D. SANTAREN, S. Sitch, N. Viovy, et al. Fluxnet and modelling the global carbon cycle. *Global Change Biology*, 13(3):610–633, 2007.

B.-C. Gao. Ndwi—a normalized difference water index for remote sensing of vegetation liquid water from space. *Remote sensing of environment*, 58(3):257–266, 1996.

T. Howell, D. Meek, and J. Hatfield. Relationship of photosynthetically active radiation to shortwave radiation in the san joaquin valley. *Agricultural Meteorology*, 28(2):157–175, 1983.

M. Jung, C. Schwalm, M. Migliavacca, S. Walther, G. Camps-Valls, S. Koirala, P. Anthoni, S. Besnard, P. Bodesheim, N. Carvalhais, et al. Scaling carbon fluxes from eddy covariance sites to globe: synthesis and evaluation of the fluxcom approach. *Biogeosciences*, 17(5):1343–1365, 2020.

C. Papagiannopoulou, D. G. Miralles, M. Demuzere, N. E. Verhoest, and W. Waegeman. Global hydro-climatic biomes identified via multitask learning. *Geoscientific Model Development*, 11(10):4139–4153, 2018.